# WHY WE NEED NEW BENCHMARKS FOR LOCAL INTRINSIC DIMENSION ESTIMATION

**Piotr Tempczyk**[*][†]
IDEAS Research Institute
Institute of Informatics, University of Warsaw
NASK - National Research Institute

**Dominik Filipiak**[‡]
Adam Mickiewicz University, Poznań
University of Innsbruck
Perelyn

**Łukasz Garncarek**
Brainly

**Ksawery Smoczyński**
Synerise

**Adam Kurpisz**[*]
BFH Bern Business School
ETH Zurich

## ABSTRACT

Neural Local Intrinsic Dimension (LID) estimators are typically bound to domain-specific architectures whose inductive biases can yield inconsistent estimates for the same underlying manifold. Existing evaluations either use overly simple synthetic data (with known LID) or real datasets (with unknown LID), obscuring true performance. We introduce a principled benchmarking framework that (i) maps the *same* manifold into multiple domain representations while preserving its structure, enabling like-for-like cross-architecture tests; (ii) designs harder variants of popular datasets that target key manifold properties; and (iii) applies controlled transformations with known LID shifts to stress-test methods even when absolute LID is unknown. Across this suite, including non-trivial synthetic datasets, we show that accuracy on simple manifolds does not transfer across domains and that state-of-the-art methods fail under targeted stressors, revealing clear failure modes and areas for improvement. Data and code are available: https://github.com/DominikFilipiak/LID-Benchmarks.

## 1 INTRODUCTION

| Algorithm / LID estimation aspect | Non-uniform density (3.1) | Manifold curvature (3.2) | Boundaries of manifolds (3.3) | Thin Manifolds (3.4) | Nearby Manifolds (3.5) | Inductive bias invariance (3.6) | Real-world dataset size dependence (3.7) | Artificially added dimensions (RWD) (3.8) | Upscaled manifold (RWD) (3.8) | Stretched manifolds (RWD) (3.8) | Synthesized real-like datasets (3.9) |
|---|---|---|---|---|---|---|---|---|---|---|---|
| **ESS** (Johnsson et al., 2014) | H | H | L | H | H | - | L | L | L | M | L |
| **NB** (Stanczuk et al., 2024) | L | L | M | L | L(M) | L | H | M | M | L | O |
| **LIDL** (Tempczyk et al., 2022) | O | O | O | O | O | L | L | L | L | L | - |
| **LIDL** (org. manifold) | M | M | H | L | L | - | - | - | - | - | - |
| **FLIPD** (Kamkari et al., 2024) | O | O | O | O | O | L(L) | L | L | L | O | L |

Table 1: Summary of our experiments. Columns: LID-estimation aspects tested by our benchmarks; rows: neural-based algorithms (ESS as a classical benchmark). Performance of methods was classified following the legend – H: high, M: moderate, L: low, O: out-of-range (unassessable) (more can be found in Sec. B). Gray color marks cells where similar aspects in the method's original paper were investigated; parentheses give that paper's reported performance (assumed H if absent). We show that algorithms that passed original simple tests for many of aspects did worse on our benchmarks; many aspects – especially on real-world datasets (RWD) – remain untested and some only partly explored.

---

[*]Equal contribution.

[†]Correspondence to piotr.tempczyk@ideas.edu.pl.

[‡]Correspondence to df@amu.edu.pl.

Recent advances in neural methods have accelerated LID estimation. A prominent line leverages generative models Tempczyk et al. (2022; 2025); Kamkari et al. (2024); Stanczuk et al. (2024); Yeats et al. (2023); Horvat and Pfister (2022; 2024). Despite the rapid development of LID estimation algorithms, scant attention has been paid to evaluating their performance. Most existing benchmarks for assessing LID estimation methods follow two common strategies. The first involves datasets sampled from well-defined and simplistic distributions where the LID is known and well-understood. The second strategy evaluates LID estimation methods on domain datasets with unknown LID.

The main advantage of the first method lies in the well-understood ground truth of LID making the evaluation of LID estimation methods reliable. Unfortunately, those datasets fail to capture the complexity of the real-world manifolds. Moreover, most existing works do not consider edge cases such as varying manifold thickness or neighboring manifolds, nor do they analyze results in a way that reveals inefficiencies of their algorithms even on simple datasets like Gaussian distributions.

On the other hand, the second approach for testing LID estimation methods, which involves using real-world domain datasets, matches the desired level of complexity but suffers from a significant drawback: in most cases, the ground truth of LID is unknown. This makes it impossible to assess a method's performance reliably, leading to possibly erroneous conclusions.

**Our contribution** Following the discussion on existing benchmarks and their limitations, we present a list of contributions to the field of testing LID estimation methods, together with the limitations that we address.

- We introduce a toolbox that maps datasets into different domain representations (e.g., images) without altering the underlying manifold, enabling controlled like-for-like tests of the same dataset across neural architectures and domains; using this toolbox, we show that the common procedure of validating on simple synthetic manifolds and silently assuming similar performance across domain networks is false.

- We introduce variants of the datasets already studied in the literature that pose much higher difficulty for modern neural-based methods (see Table 1). While many of the presented methods have been reported to achieve high accuracy, we show that a more careful design of datasets targeting key manifold characteristics poses significant challenges for LID estimation methods.

- We bridge the gap between benchmarks based on well-understood analytical distributions and real-world datasets by employing several data transformations. As a result, we are able to stress-test algorithms on datasets with unknown LID by evaluating their performance before and after transformation, and by comparing it to the ground-truth LID difference imposed by the transformations. We also show how drastically sample size for real-world datasets affects tested algorithm estimates.

- We show the significance of testing the algorithms on non-trivial synthetic datasets, that pose much harder challenge than other types described before.

- Finally, the majority of modern methods are tested on different datasets, which hinders understanding of the limitations that should fuel further development. This paper provides a broader collection of datasets, allowing more meaningful analysis and comparison across various methods (see Table 1).

In the fast-moving field of LID estimation, we believe that the results presented in this paper shed new light on the construction of testing benchmarks and can amplify the impact of this work by informing and supporting the many new techniques currently being explored in research labs around the world. In particular, authors of existing methods, especially those analyzed in our study, may find this paper helpful in refining or extending their own approaches. Finally, this work also highlights the need for further development of benchmarks to keep pace with the rapid and diverse advancements in LID estimation techniques. We overview LID applications, evaluated algorithms, and related work in Sec. C and D. Practical guidance on LID estimation algorithms can be found in Sec. G.

## 2 METHODS FOR CREATING DOMAIN DATASETS

We use a set of transformations and methods that can be used to create challenging benchmarks. While they in principle can work for any continuous domain (such as audio, video, EEG, etc.), we focus on the images domain in this work.

**Inverse Domain Representation (IDR)** The goal of this method is to bridge the gap between datasets sampled from analytical distributions with known ground truth of LID and real-world datasets on arbitrary domains. While producing an artificial dataset sampled from an arbitrary manifold is not challenging (one can embed the manifold in $\mathbb{R}^D$ and sample from the embedding's range), this becomes more complex when one wants the dataset to resemble some real-world data. We need a method to embed an arbitrary manifold into the ambient space of a given dataset $X \subset \mathbb{R}^D$ in a manner such that the image is "close to" $X$. For a formal definition of IDR and a detailed discussion, we refer the reader to Appendix E. For the purpose of experiments class 7 images from FMNIST were used as a basis for IDR transformation, samples for such dataset are presented in the Figure 1

**Monotonic Embedding (ME)** The goal of this method is to assess the robustness of LID estimation algorithms to continuously differentiable geometric deformations of the data manifold. It is particularly useful for datasets with unknown intrinsic dimensionality. The procedure involves applying continuously differentiable, monotonic transformations to the coordinates of the data in the ambient space, effectively stretching or compressing the geometry in a controlled manner. Notably, different functions may be applied independently to each coordinate, allowing for highly flexible distortions. As long as the derivative of the applied function remains controlled, we expect the LID estimates to remain stable before and after the transformation. This follows from general theory—it is a fundamental result that diffeomorphisms preserve manifold dimension, and the mappings we use are diffeomorphisms.

In the other methods presented, we can similarly assert that what we actually do is applying a sufficiently regular mapping to our manifold.

**Ambient Space Extension (ASE)** This method alters the ambient dimensionality of a dataset without modifying the LID. Similarly to the previous approach, it is suitable for datasets with unknown LID and can be used to confirm algorithm stability on a dataset. The extension is performed by introducing new dimensions as deterministic, continuously differentiable, and monotonic functions of the original coordinates. In the case of image data, this could be achieved through deterministic upscaling techniques. More formally, in ASE we re-embed $M$ using a map of the form $x \mapsto (x, F(x))$ for $x \in \mathbb{R}^d$, which has a smooth inverse, namely the projection onto the first $d$ coordinates, and thus preserves the dimension. At first glance, such modifications may appear trivial. However, when viewed through the lens of deep learning models, especially convolutional neural networks, they can introduce significant complexity. The hierarchical structure of learned convolutional features may differ considerably between models trained on original versus extended datasets.

**Auxiliary Dimension Injection (ADI)** This method increases the ambient dimensionality of datasets with unknown LID by adding informative features derived from parametric transformations of the original data. Parameters are sampled from known distributions, ensuring the result remains structurally related to the source dataset. Notably, we apply maps that have non-degenerate Jacobians, so they are immersions and preserve the local dimensions of their domains. The increase of dimension is determined by the dimension of the co-domain of the new embedding we define. The approach is highly flexible. For example, an audio signal may be filtered with a random low-pass cutoff and blended with the original at a random ratio, yielding two additional, non-trivial dimensions. In the image domain, concatenating random pairs of MNIST digits produces samples whose dimensionality equals the sum of the individual image dimensions, as the new dataset effectively forms a Cartesian product of the original space with itself.

**Manifold Synthesis (MS)** We generate datasets with known intrinsic dimensionality yet complex, non-trivial geometry by applying deterministic, continuous transformations to a parameterized manifold that preserve topological and differential properties. For example for images, one can parametrize object attributes (e.g., position, orientation) and render; for audio, one can combine sample fragments via controlled parameters (start time, filter cutoff, duration, volume). The result is data with well-defined LID but with substantially more complex appearance and structure.

## 3 Algorithm analysis demonstrated using image datasets

In this section, we use methods presented in related work in Sec. 2 to create datasets designed to test various interesting aspects of LID estimation algorithms, along with a discussion of their construction and characteristics. The details of experimental setting can be found in Sec. F. Moreover, for the introduced datasets, we present the estimated LID for the tested algorithms. We focus on presenting one plot of interesting results for selected (often good-performing) algorithms, while the remaining ones are shown in Sec. B. All the results are summarized in tables in Sec. A.

### 3.1 Non-uniform densities

As experiments Tempczyk et al. (2022); Stanczuk et al. (2024); Kamkari et al. (2024) and theoretical considerations Tempczyk et al. (2025) show, LID estimation for non-uniform densities may be close to correct value when averaged on the whole dataset but biased in certain areas, e.g., in LIDL this bias is a function of the Laplacian of the density.

For example, in LIDL, the core estimate (Theorem 2.1) expresses the logarithm of the diffused density as a sum of a linear term (whose slope allows to recover the dimension) and an error term (the effect of which diminishes as $\delta \to 0$, but affects the final estimation as in practice we probe with concrete values of $\delta > 0$). The magnitude of the error term can be traced through the proof of the core estimate to arise from non-constancy of density and curvature of the manifold. Therefore, it is necessary to more thoroughly examine the behavior under non-uniform densities.

**Gaussians (IDR)** The dataset is a mixture of four 5-dimensional Gaussian distributions with means located at $(-3, 3, 0, 0, 0)$, $(3, -3, 0, 0, 0)$, $(-3, -3, 0, 0, 0)$, $(3, 3, 0, 0, 0)$ with standard deviations $1/27$, $1/9$, $1/3$, and $1$ respectively, transformed with the IDR method. For each point, we calculated the distance from the closest mean and divided this distance by the appropriate standard deviation. Results for each algorithm are presented in Figure 3 and 14. We would expect that no matter from which component of the mixture we sample and no matter how far we are from the mixture component mode, the estimate should be equal 5. Only ESS achieved satisfying results on this task.

### 3.2 Manifold curvature

Building on the motivation from the previous section and the fact that the majority of algorithms are tailored for flat, uniform manifolds, we aim to test the behavior of LID estimation algorithms on curved manifolds. Curved manifolds are commonly present in real-world scenarios.

**Spheres (IDR)** We used 4 disjoint spheres $S^4$ with origins located in $(-3, 3, 0, 0, 0)$, $(3, -3, 0, 0, 0)$, $(-3, -3, 0, 0, 0)$, $(3, 3, 0, 0, 0)$ and with radia of $1/27$, $1/9$, $1/3$, $1$ respectively, transformed with the IDR method. For each point in the dataset, we calculated the distance from the four sphere origins and assigned those points to the closest sphere. Estimate distributions for each algorithm and each sphere separately are presented in Figure 4 and 15. The only algorithm unaffected by the curvature of this dataset was ESS.

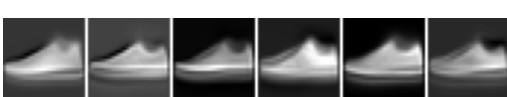

Figure 1: Few samples from Gaussian (IDR) dataset.

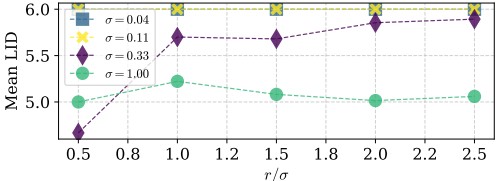

Figure 3: LID estimates calculated using NB. This dataset – Gaussians (IDR) – is composed of a mixture of four 5-dimensional Gaussians. Each line represents the average LID estimate as a function of a standardized distance of a point from its corresponding component mean.

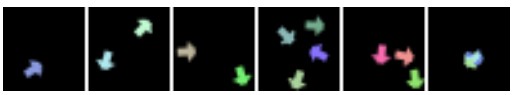

Figure 2: Sample from Arrows (MS) dataset.

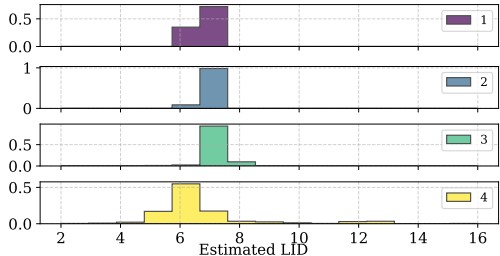 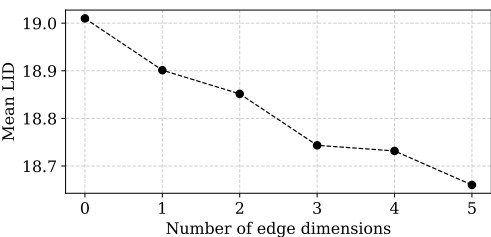

Figure 4: LID estimate distribution computed using NB for the Sphere (IDR) dataset composed of four disjoint spheres of different radii. Numbers from 1 to 4 indicate the sphere number order from smallest to largest sphere.

Figure 5: LID as a function of edge dimensions for ESS on 20-dimensional Uniform (IDR) distribution between $-3$ and $3$.

**Spaghetti (IDR)** The dataset used in this analysis is the spaghetti line dataset introduced by Stanczuk et al. (2024) but transformed into an image domain using IDR. It is a 1-dimensional manifold homeomorphic with the circle twisted and folded that it occupies $k = 20$ dimensions. Points from this manifold are sampled as follows: $\theta \sim \mathcal{U}(0, 2\pi)$; $x_i = sin((i+1)\theta)$, for $i = 1, \ldots, k$. Results are presented in Table 2 and reveal that this dataset can pose a serious challenge for some algorithms. Notably, as shown by Stanczuk et al. (2024), the NB algorithm could solve the dataset without IDR domain transformation up to an embedding into $k = 100$ dimensions with high accuracy. However, the IDR transformation introduces significant challenges, leading to higher (but still reasonable) errors even for $k = 20$.

### 3.3 Boundaries of manifolds

An extreme case of non-uniform density is a distribution with sharp edges, like a uniform distribution on a hypercube. Such distributions are common in real-world datasets due to measurement limitations. For example, cameras can't record light intensity beyond a threshold, so image datasets often lie within a hypercube, with many points on its boundary (any images with some white or black pixels).

Many algorithms work under the assumption that when measuring LID at a point $x$, we are considering a sufficiently small neighborhood of $x$ where density has some *nice* properties, e.g., being sufficiently smooth. However, in practice, the neighborhood under consideration is contained in a ball of some radius $r$, whose center can lie in the proximity of a boundary or precisely on it. This leads to a problematic case, which was observed empirically by Tempczyk et al. (2022), and formalized later by Tempczyk et al. (2025).

**Uniform (IDR)** The dataset is a 20-dimensional uniform distribution between $-3$ and $3$ on each dimension, transformed with the IDR method. In this test, we are grouping points that are in the proximity of $m$ edges. A point is said to be close to an edge if it lies on the original manifold closer than $0.25$ from the edge located at $3$ or $-3$. We test LID estimation for various values of parameter $m$. Results are presented in Figure 5 and 16. In the first figure, a monotonic relationship between the average LID and a number of edge dimensions for the ESS algorithm can be observed. The pattern is less visible for other algorithms though.

### 3.4 Thin manifolds

An interesting scenario arises when moving along the manifold on a certain path, where the local intrinsic dimension remains unchanged. However, in an orthogonal direction to this movement, the manifold becomes thinner. In such cases, we would expect to observe consistent LID estimations across all observations in a reasonable range. Nevertheless, some algorithms might mistakenly identify the manifold as having a lower local dimensionality.

**Moon (IDR)** The manifold we study is 3 dimensional. It is moon-shaped in the first two dimensions and a uniform interval in the third one, parameterized by $z$. More formally, it is sampled uniformly at

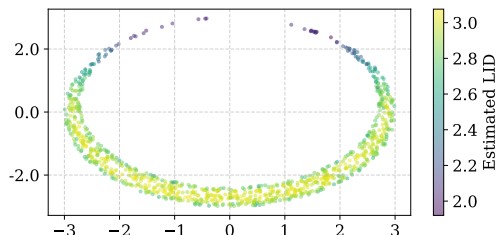 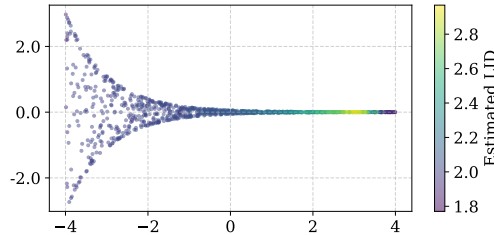

Figure 6: LID estimates for the Moon (IDR) dataset using ESS.

Figure 7: LID estimates for the Funnel (IDR) dataset using ESS.

random from the points $(x_1, x_2, x_3) \in \mathbb{R}^3$ intersected with the set $\mathcal{M}$ defined as

$$\mathcal{M} \coloneqq \{\|(x_1, x_2)\| \leq r, \ \|(x_1, x_2 + 0.1)\| \geq 0.899r, |x_3| \leq z\},$$

where $r$ is a radius hyperparameter that can be chosen arbitrarily. For our experiments, we used the value $r = 3$. The resulting manifold has a thickness of $0.201r$ on the bottom part and $0.001r$ on the upper part. Finally, the dataset is transformed using the IDR method.

Figure 6 and 17 show results for different algorithms. ESS has the best performance among them. It was able to maintain the correct estimate until the manifold was very thin; on the other hand, the NB estimates were close to ground truth but with errors distributed quite independently of manifold thickness. What is interesting is that we can observe a slight drop in the estimate for ESS close to the border of the moon, which is the same effect that was observed on the edge of uniform distribution.

### 3.5 NEARBY MANIFOLDS

The parallel 1D manifolds example in Tempczyk et al. (2022; 2025) suggests, that when the expected euclidean distance in the ambient space to the nearest neighbor from other region of the manifold (or other manifold) is similar or smaller to the expected distance to the local neighbor on the same region of the manifold, it may lead to a bias in the estimate. The ability to recognize separate manifolds close to each other for finite sample size is a desirable property for LID estimation algorithms. To showcase this phenomenon, we introduce the Funnel and Spiral datasets.

**Funnel (IDR)** We consider a 2-dimensional funnel embedded in 3 dimensions, visualized in Figure 7. The first two coordinates of the manifold original space: $x_1$ and $x_2$, and generated using this set of equations and transformed with the IDR method:

$$t = \mathcal{U}(0, 8); \ r = 3 \exp(-t); \ \theta = \mathcal{U}(0, 2\pi);$$

$$x_1 = t - 4; \ x_2 = r \sin \theta; \ x_3 = r \cos \theta.$$

Results are presented in Figure 7 and 18. We can observe that even when the algorithm gives a proper estimate when the radius of the funnel is high, for a low radius, the estimate is distorted. The ESS algorithm behaves as expected. For wide parts of the funnel, the estimate is exact. For the narrower parts, it goes up because manifolds are close to each other, and the algorithms start to detect points in all 3 dimensions. On the right end, it goes down when it starts to resemble a 1-dimensional line. The rest of the algorithms give more or less biased and noisy estimates compared to ESS.

**Spiral (IDR)** A data set we consider is a spiral dataset visualized in Figure 8 for the first two coordinates $x_1$ and $x_2$, generated using the set of equations:

$$t = \mathcal{U}(1, 100); \ r = 1/t; x_1 = r \sin(t/r); \ x_2 = r \cos(t/r),$$

where the distance to the closest point from the next revolution of the spiral gets smaller when we go down the spiral. The dataset has a useful property: the expected distance to the nearest neighbor calculated on the manifold is the same at any point on the manifold. Such a constructed dataset is then transformed with the IDR method.

An interesting observation for ESS algorithm, showcased in Figure 8, is that the algorithm shows an LID estimate close to 2 only during the first revolution, where the manifold visually has an LID equal to 1 for a few more revolutions, especially when looking at full dataset which is $100\times$ bigger

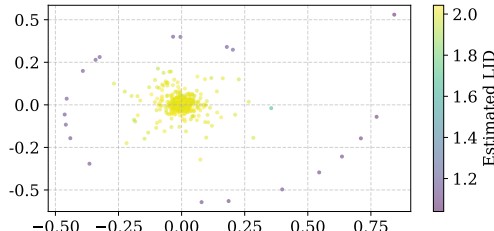 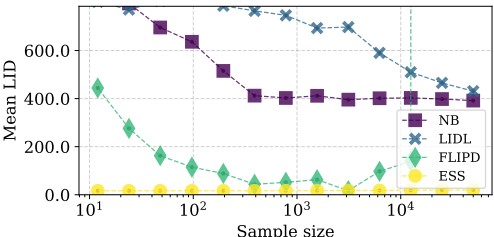

Figure 8: LID estimates for the Spiral (IDR) dataset using ESS. The test set shown in the figure is 100× less dense than the training set.

Figure 9: LID estimate in sample size on FM-NIST. The x-axis is shown in logarithmic scale to highlight differences in the small-sample regime (n < 100),

than the test set. We want to highlight the fact that the algorithm was run with a hyperparameter of 100 neighboring points, which is a standard value for this parameter. We observed in our experiments that reducing the parameter responsible for the number of neighbors in ESS can reduce the estimate error by giving correct estimates further down the spiral. However, this dataset dependent hyperparametrization is a problematic approach for datasets with unknown LID that cannot be inspected visually. We also note that the ESS algorithm performed very well compared to the remaining methods, for which the results are presented in Figure 19.

## 3.6 LACK OF NETWORK ARCHITECTURE INVARIANCE

In our experiments we observed, that for LIDL and FLIPD use of convolutional networks in the algorithm (Glow(Kingma and Dhariwal, 2018) and U-net(Ronneberger et al., 2015) respectively) yields worse results than using feedforward based networks (MAF(Papamakarios et al., 2017) and MLP) on the same manifold but transformed using IDR. We can see in Table 4 that some of the effect may be just from widening the ambient space, but similar effect were reported in Kamkari et al. (2024). Our experiments for NB show that on Gaussian and Sphagetti datasets transformed by IDR we obtain different results than in the original paper without IDR transformation (more in Sec.3.2).

## 3.7 ESTIMATED LID VS SAMPLE SIZE

Tempczyk et al. (2022) showed that for numerous algorithms the bias of the estimate is dependent on sample size. While it is natural that an algorithm's error gets smaller for bigger sample sizes, an introduced bias leads to a situation where we don't know if we have enough data for our estimate to be correct. To test that we created a series of training datasets by drawing samples of different sizes from the FMNIST dataset. The validation set used for early stopping and the test set was held the same. One may wonder why we chose a real-world dataset rather than an artificial one with known LID. Figure 8 in Tempczyk et al. (2022) shows that such a dependency does not occur for LIDL on artificial data, so we aimed for more challenging dataset as our experiments demonstrated that LIDL exhibited a noticeable bias for small sample sizes on the FMNIST dataset.

The results are presented in Figure 9 and 20. We observe a rather significant dependence of the estimate on the sample size. Among all the algorithms, the NB algorithm achieves interesting and desirable characteristics. It stabilizes average estimate values for datasets bigger than 1000 samples, which is a good result compared to other algorithms. One interesting algorithm in this context is Erba et al. (2019), which is designed to deal with undersampled regions, but we did not test it in our work.

## 3.8 REAL-WORLD DATASET TRANSFORMATIONS

For most real-world datasets used to evaluate the performance of LID estimation methods, the ground truth of LID is unknown. The absence of ground truth makes it impossible to reliably assess the quality of LID estimates produced by these algorithms in such domains. In what follows, we propose a set of transformations on such datasets that modify dimensionality in a controlled manner, enabling a rigorous evaluation of algorithm performance by measuring the difference in LID before and after applying the transformation. In all experiments in this subsection, we used a downscaled version of

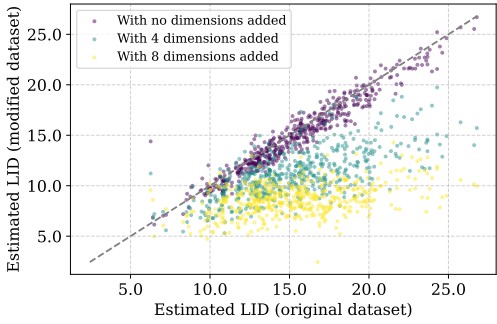 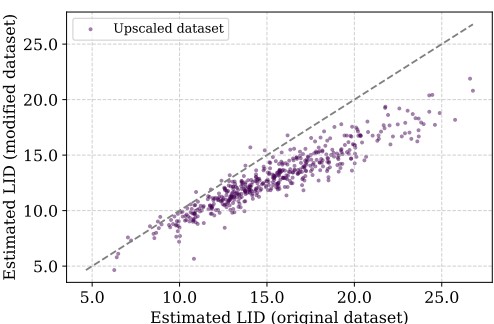

Figure 10: LID estimated on modified dataset minus the added dimensions versus the LID estimate on the original data set, the FMNIST 16×16, for the ESS algorithm. The number of added dimensions in the legend.

Figure 11: LID before upscaling vs LID after upscaling of the FMNIST images for ESS algorithm.

FMNIST as our base dataset. The original images were resized to 16×16 pixels. Results from this section are presented in Table 3.

**Added dimensions (ADI)** In this dataset, we added an 8-pixel-wide frame using mirror padding in eight directions, creating images of size 32×32 pixels. We generated three variants of the dataset with 0, 4, or 8 added dimensions by applying random brightness changes to a respective number of reflections. The results of LID estimation for the ESS algorithm are presented in Figure 10. The estimates for datasets with added dimensions were calibrated by subtracting the number of added dimensions from the respective estimates, ensuring that the results should align with the identity line.

For ESS, points where LID estimation indicated a lower dimensionality in the original dataset remain close to the identity line, suggesting a small relative error after image transformation. In contrast, for points where LID estimation in the original dataset was high, the estimates after transformation fall significantly below the identity line, indicating large estimation errors. The LIDL algorithm, in turn, adds dimensions even when the dataset has zero additional noisy reflections, meaning no additional dimensions were introduced. NB maintains the estimate on the modified dataset close to the identity line for some points, but for others, it overestimates the LID regardless of the number of added noisy reflections. Another observation is that the estimates vary significantly, sometimes by more than ±50% compared to the same estimate on the base dataset. All results are presented in Figure 21.

**Upscaled (ASE)** The dataset used was an upscaled 32×32 version of the base dataset. For upscaling, we used a torch function interpolate with bilinear mode. NB's performance in this task is quite impressive, despite the variability of the estimate, which remains of a similar magnitude as in the previous experiment involving added dimensions. ESS returns estimates lower than the original ones, while FLIPD and LIDL exhibit the opposite behavior, outputting higher estimates. All results are presented in Figure 11 and 22.

**Stretched (ME)** We create a ME by using a polynomial transformation applied to each pixel after normalizing its values to the range $[0, 1]$. We performed two transformations using different exponents: $x \in [0, 1] \mapsto y = x^l$, for $l \in \{0.25, 4\}$. The best-performing algorithm in this case is ESS, which maintains a similar mean LID before and after the transformation, albeit with a high variance. NB and LIDL estimates drop significantly after the spatial transformation, while FLIPD heavily overestimates the LID after transformation. Full results in Figure 12 and 23.

## 3.9 REAL-LIKE DATASET WITH KNOWN LID

There are almost no datasets that simultaneously resemble real-world images and have a known underlying LID. Two notable exceptions are the Gaussian blobs from Stanczuk et al. (2024) and 3DIdent dataset introduced by Zimmermann et al. (2021). The latter one was too big for our computational budget and available GPU resources. Therefore, we created similar dataset using MS approach.

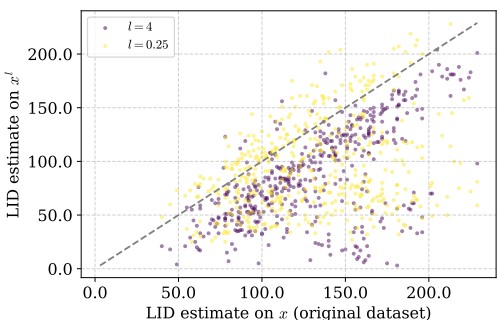

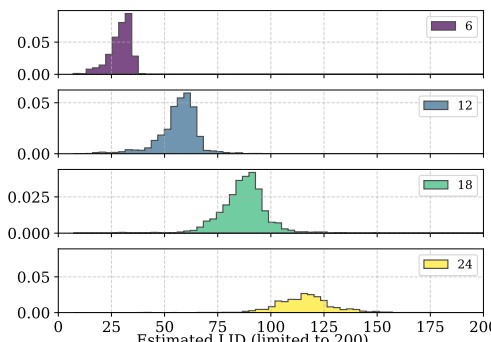

Figure 12: LID estimates for NB on FMNIST before and after the transformation $x^\ell$, with two values of $l \in \{0.25, 4\}$.

Figure 13: LID distribution estimated using NB for different manifolds in arrows dataset. Manifold dimensionality in legend.

**Arrows (MS)** The dataset consists of $32 \times 32$ images of arrows placed on a black background. Each arrow is described by six variables: horizontal and vertical position, rotation, and color (three variables for RGB). The manifold dimensionality is six times the number of arrows in the image. A sample is shown in Figure 2. There is a small possibility of manifold collapse when arrows perfectly overlap, but the probability of this occurring is less than $10^{-3}$, making it negligible.

In this experiment, none of the algorithms produced results close to the ground truth. The ESS estimate failed to distinguish between different manifolds, consistently outputting values around 15 for all cases, with variations appearing only in an uncontrolled manner. The NB algorithm significantly overestimated LID values and produced some outlier estimates for LID of a value around 3k. FLIPD produced estimates ranging from -20 to 40. Figures 13 and 24 present full results. We comment this results in Sec. G.

## 4    ANALYSIS OF ALGORITHMS PERFORMANCE

**ESS** The ESS algorithm performed very well for datasets with low-dimensional manifolds. From experiments of Tempczyk et al. (2022) we know, that its performance deteriorates if manifold dimensionality raises, which can be observed on 20D uniform datasets. We observed in preliminary experiments that its behavior can change when working on smaller samples, and the `n_neighbours` parameter is crucial to the performance, but there is no prior way of setting it right. We could go with the highest value possible due to the computational and memory constraints, but in the case of the Spiral dataset the smaller values works better, especially for smaller sample sizes like $10K$, so there is no right answer to that problem.

**NB** Performed well, especially compared to other algorithms using neural networks. It failed on arrows, stretch, gaussian and spheres and had higher error than ESS on low-dimensional manifolds. It will surely beat ESS on higher-dimensional manifolds due to the ESS underestimation bias for higher-dimensions, but compared to ESS it lacks the precision and robustness on simpler manifolds. This perhaps may be corrected by the person skilled in training diffusion models, but in practice this method will benefit from more stable and less noisy estimates, especially because some tests were failed due to the high variance and not high bias like in the case of other methods like LIDL and FLIPD.

**LIDL** This algorithm performed much worse than expected based on its performance on datasets with known LID from the original paper. In the first experiments, we used LIDL with Glow Normalizing Flow Kingma and Dhariwal (2018), but it performed worse than MAF Papamakarios et al. (2017), and its training time was an order of magnitude slower than MAF so finally we sticked with MAF. To further investigate this algorithm, we ran LIDL on selected datasets in three different scenarios: before the IDR transformation, before the IDR transformation but padded with zeros to reach 784 ambient space dimensions, and after the IDR transformation. This analysis provided deeper insight into where the accuracy of LIDL's estimates deteriorates. In the first case, where the original ambient space before IDR was 30-dimensional, LIDL performed well. However, the IDR transformation

or expanding the ambient space caused LIDL's performance to deteriorate. The results of these experiments are presented in Table 4.

Yet there is another problem with LIDL: although theoretical results in Tempczyk et al. (2025) show that when $\delta \to 0$ we should get an unbiased estimate, in practice we always get the estimate close to ambient space dimensionality. This is an unsolved problem of how to choose this parameter in real-world scenarios. Those results are presented in Sec. H, where we can observe how much the LIDL estimate varies with $\delta$. For presenting the results we have chosen one $\delta$ range which had the best performance on IDR datasets, still not being even close to the underlying LID value.

**FLIPD** This algorithm suffered from the same problems as LIDL, but to a greater extent. Authors of the FLIPD describe a "knee" on the plot where the estimate is the closest to the ground truth, but in practice, those models many times had problems converging and producing unreliable estimates with a hard-to-find "knee" structure. When we knew the underlying LID we were able to present better results when choosing the value of $t$ with smaller MAE, but it is not a practical scenario for real-world datasets. The results for different values of $t$ are presented in Sec. H, where we can observe how much FLIPD estimate varies with $t$.

Authors of Tempczyk et al. (2025) show that the theoretical foundations behind LIDL and FLIPD are solid and give tools to calculate reasonable ranges of delta for the given problem, so it suggests that the problem is the non-ideal density estimator.

## 5 CONCLUSIONS AND FUTURE WORK

This paper shows that there are many aspects of LID estimation that should be investigated while working on new algorithms in the future. The results strongly suggest that the domain-adapted benchmarks are a crucial aspect of this process: very similar algorithms in terms of performance on the classical benchmarks turned out to be very different when tested in a new way. There are many aspects that are not covered in this work and should be further investigated, like the effects of quantization of the dataset on LID estimate, testing on higher-dimensional manifolds, and dealing with the noise in the data. There is also a need to test similar algorithms in other domains, such as audio and other real-world datasets similar to the arrows of 3DIdent. We hope that our work will be an inspiration for future research and will aid progress in the area of LID estimation.

Although we provide detailed, plot-based evaluations of each aspect and method, as methods improve there will be a need to quantify each aspect with some metric in a principled way. But we believe that we are not there yet and current algorithms still require refinement to the point at which majority of scores reach at least M in Table 1.

**Acknowledgements.** This work was partially supported by Sonata Bis grant 2024/54/E/ST6/00388 awarded by NCN Poland.

**CRediT Author Statement. Piotr Tempczyk** (29% of the work): Conceptualization, Methodology, Software, Validation, Formal analysis, Investigation, Data Curation, Writing – Original Draft, Writing – Review & Editing, Visualization, Supervision, Project administration. **Dominik Filipiak** (28% of the work): Investigation, Software, Data Curation, Writing – Review & Editing, Visualization. **Łukasz Garncarek** (9% of the work): Conceptualization, Methodology, Formal analysis, Investigation, Writing – Original Draft, Writing – Review & Editing. **Ksawery Smoczyński** (5% of the work): Software, Data Curation, Writing – Review & Editing. **Adam Kurpisz** (29% of the work): Conceptualization, Methodology, Validation, Formal analysis, Investigation, Writing – Original Draft, Writing – Review & Editing, Visualization, Supervision.

**Reproducibility Statement.** We provide the dataset along with source code and scripts required to generate it on GitHub: `https://github.com/DominikFilipiak/LID-Benchmarks`. As for LID algorithms, implementation details and hyperparameters are described in Appendix F.

**Ethics Statement.** This work concerns methodology for Local Intrinsic Dimension (LID) estimation, a Topological data-analysis problem. It does not involve human subjects, personally identifiable information, or sensitive attributes, and it introduces no foreseeable direct societal harms. All datasets used are public or synthetic, and we follow their licenses and terms of use.

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

# APPENDIX

## A TABLES

Table 2: LID estimations with MAE for the datasets with known dimensionality.

| algorithm | ESS | | | FLIPD | | | LIDL | | | NB | | |
|---|---|---|---|---|---|---|---|---|---|---|---|---|
| | $\mu$ | $\sigma$ | MAE | $\mu$ | $\sigma$ | MAE | $\mu$ | $\sigma$ | MAE | $\mu$ | $\sigma$ | MAE |
| Gaussians | 4.99 | 0.09 | 0.07 | 3.25 | 3.57 | 3.08 | 17.24 | 3.09 | 12.24 | 5.72 | 0.66 | 0.81 |
| Spheres | 5.07 | 0.08 | 0.09 | 4.49 | 3.77 | 3.24 | 17.90 | 3.77 | 12.90 | 6.80 | 1.06 | 1.83 |
| Spaghetti | 1.03 | 0.03 | 0.03 | 1.12 | 4.19 | 3.54 | 10.25 | 6.40 | 9.53 | 2.12 | 1.54 | 1.12 |
| Uniform | 18.87 | 0.43 | 1.13 | 18.08 | 3.30 | 2.80 | 34.70 | 3.61 | 14.70 | 19.41 | 1.78 | 1.07 |
| Moon | 2.86 | 0.21 | 0.15 | 2.14 | 3.96 | 3.42 | 12.31 | 5.45 | 9.31 | 2.75 | 0.46 | 0.25 |
| Funnel | 2.20 | 0.26 | 0.21 | 1.73 | 3.46 | 3.24 | 14.18 | 3.09 | 12.20 | 1.48 | 0.51 | 0.54 |
| Spiral | 1.99 | 0.13 | 0.99 | 1.47 | 3.36 | 3.34 | 1.98 | 5.38 | 2.26 | 2.03 | 0.73 | 1.03 |
| Arrows | 14.73 | 3.88 | 6.23 | 8.57 | 25.80 | 17.43 | – | – | – | 455.85 | 1,007.53 | 440.82 |

Table 3: LID estimations for the modified real-world datasets with unknown dimensionality.

| algorithm | ESS | | FLIPD | | LIDL | | NB | |
|---|---|---|---|---|---|---|---|---|
| | $\mu$ | $\sigma$ | $\mu$ | $\sigma$ | $\mu$ | $\sigma$ | $\mu$ | $\sigma$ |
| FMNIST (base) | 15.32 | 3.75 | 55.92 | 24.62 | 138.73 | 44.11 | 133.45 | 38.34 |
| FMNIST (add dim +0d) | 14.87 | 3.70 | 210.03 | 45.92 | 227.01 | 62.37 | 142.40 | 37.57 |
| FMNIST (add dim, +4d) | 15.21 | 2.40 | 166.43 | 45.91 | 235.48 | 67.07 | 144.89 | 37.68 |
| FMNIST (add dim, +8d) | 16.72 | 1.65 | 193.58 | 38.85 | 257.39 | 61.89 | 154.77 | 38.91 |
| FMNIST (upscaled) | 12.86 | 2.67 | 129.57 | 38.12 | 197.13 | 65.71 | 132.78 | 39.44 |
| FMNIST (stretched $x^{0.25}$) | 14.92 | 4.44 | 49.96 | 23.21 | 105.17 | 42.25 | 98.10 | 58.18 |
| FMNIST (stretched $x^4$) | 17.98 | 4.85 | 86,355.13 | 94,294.77 | 81.68 | 40.36 | 85.74 | 43.34 |

Table 4: LID estimations with MAE for the datasets with known dimensionality.

| algorithm | LIDL (w/ IDR) | | | LIDL (org. manifold) | | | LIDL (org. manifold + padding) | | |
|---|---|---|---|---|---|---|---|---|---|
| | $\mu$ | $\sigma$ | MAE | $\mu$ | $\sigma$ | MAE | $\mu$ | $\sigma$ | MAE |
| Gaussians | 17.24 | 3.09 | 12.24 | 4.34 | 1.17 | 0.90 | 18.68 | 19.38 | 22.14 |
| Spheres | 17.90 | 3.77 | 12.90 | 4.69 | 1.10 | 0.83 | 27.49 | 12.99 | 23.28 |
| Spaghetti | 10.25 | 6.40 | 9.53 | 5.30 | 1.94 | 4.31 | 36.25 | 11.41 | 35.25 |
| Uniform | 34.70 | 3.61 | 14.70 | 19.57 | 1.06 | 0.88 | 32.46 | 10.59 | 13.58 |
| Moon | 12.31 | 5.45 | 9.31 | 2.97 | 0.31 | 0.24 | -21.31 | 18.99 | 25.57 |
| Funnel | 14.18 | 3.09 | 12.20 | 1.87 | 0.61 | 0.54 | 11.20 | 5.98 | 9.56 |
| Spiral | 1.98 | 5.38 | 2.26 | 0.85 | 0.88 | 0.66 | – | – | – |

# B ADDITIONAL RESULTS

## B.1 NON-UNIFORM DENSITIES

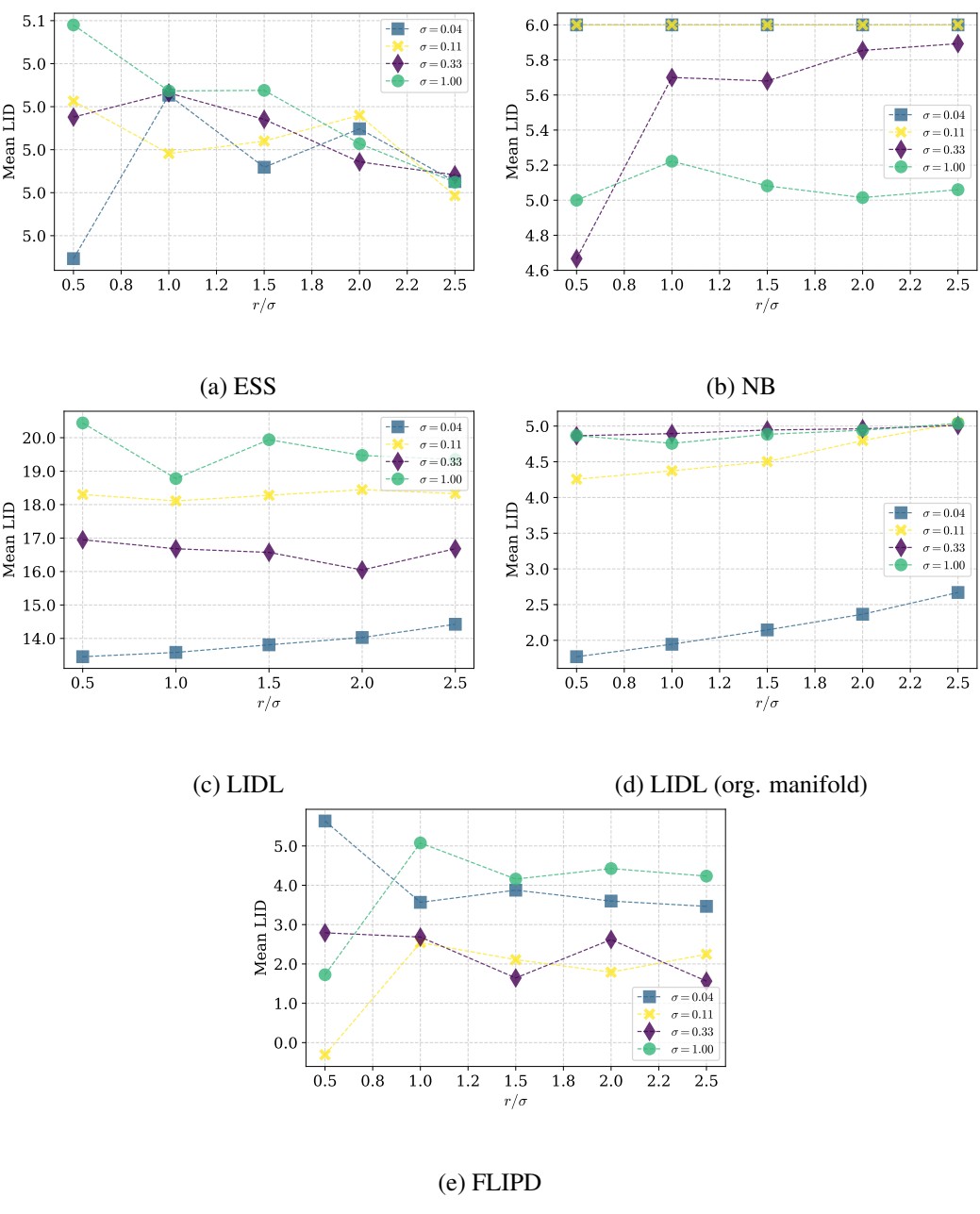

Figure 14: Results for gaussians dataset for different algorithms. ESS ranked H because estimate is almost perfect; NB ranked L because it overestimates for most of the cases; LIDL ran on original manifold yields accurate estimates for more than half of the cases, so we ranked it M.

## B.2 MANIFOLD CURVATURE

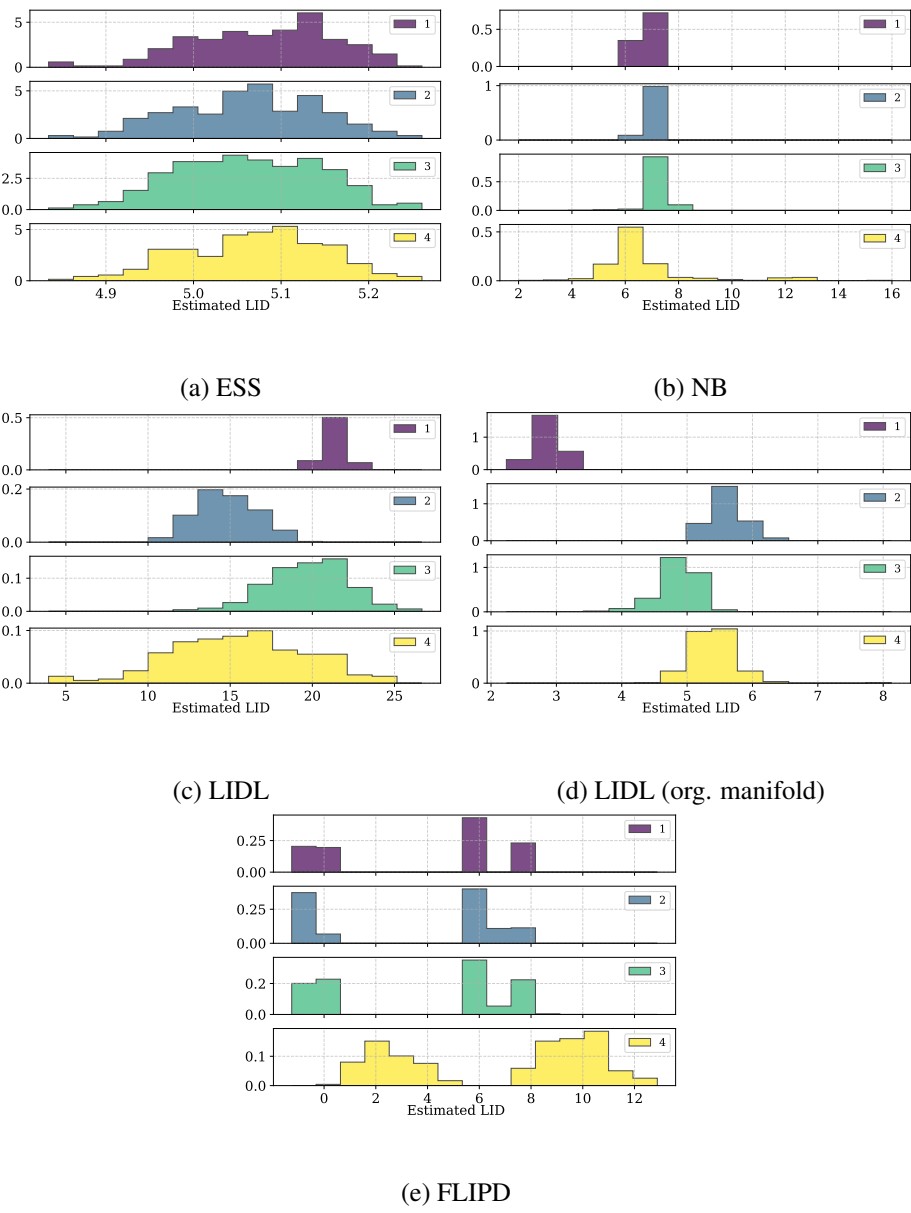

(a) ESS

(b) NB

(c) LIDL

(d) LIDL (org. manifold)

(e) FLIPD

Figure 15: Results for spheres dataset for different algorithms. ESS ranked H because estimate is almost perfect; NB ranked L because it overestimates for all of the cases; LIDL ran on original manifold yields accurate estimates for around 30% of the cases, so we ranked it M.

## B.3 BOUNDARIES OF MANIFOLDS

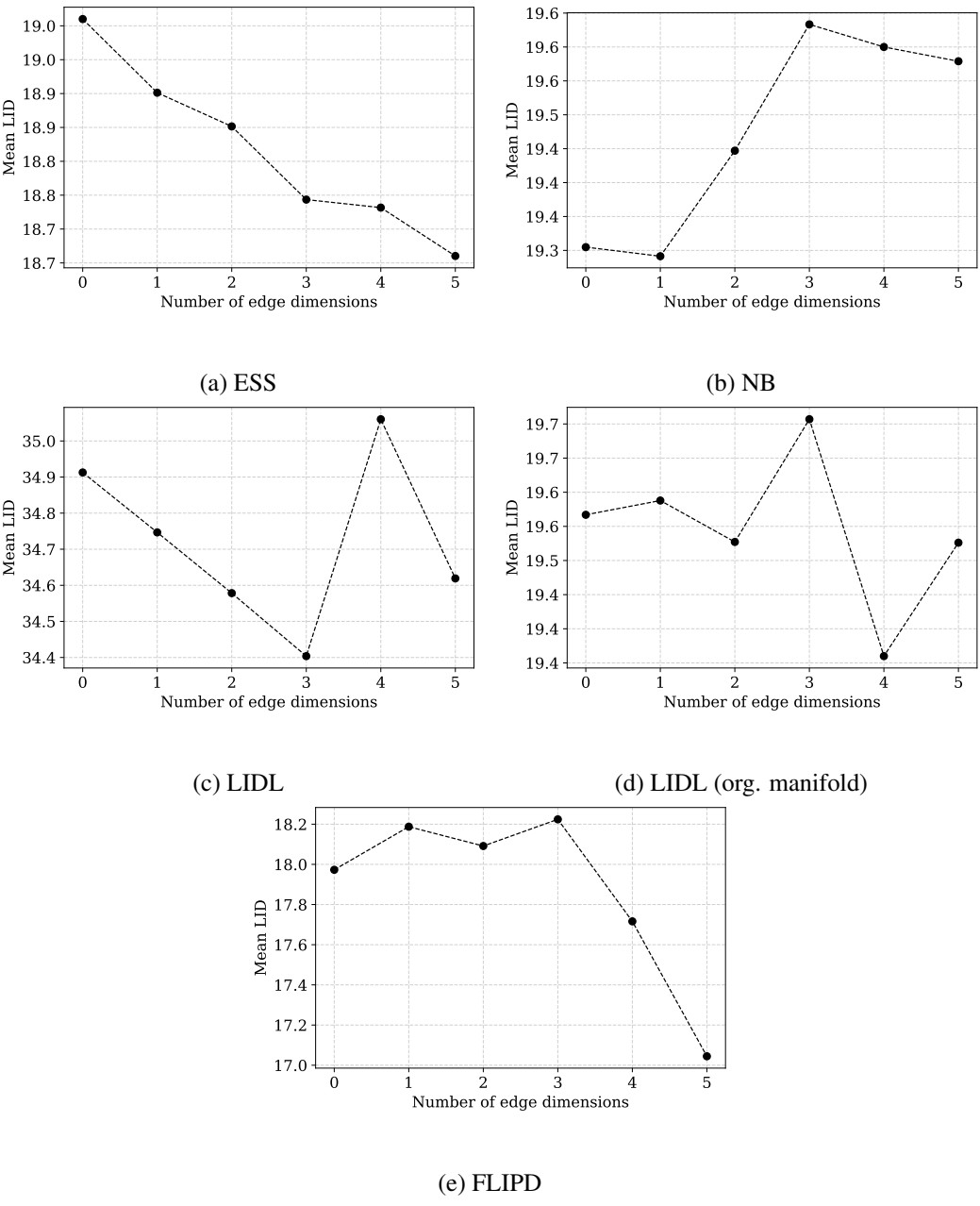

(a) ESS

(b) NB

(c) LIDL

(d) LIDL (org. manifold)

(e) FLIPD

Figure 16: Results for edge points from uniform dataset for different algorithms. ESS underestimates the true value in all cases, so we ranked it L; NB has better estimate when closer to the edge, but for most of the points further from edges it underestimates so we ranked it M; LIDL on original manifold underperformed only in cases, where points are close to 4 or more edges, so we ranked it H.

## B.4 Thin manifolds

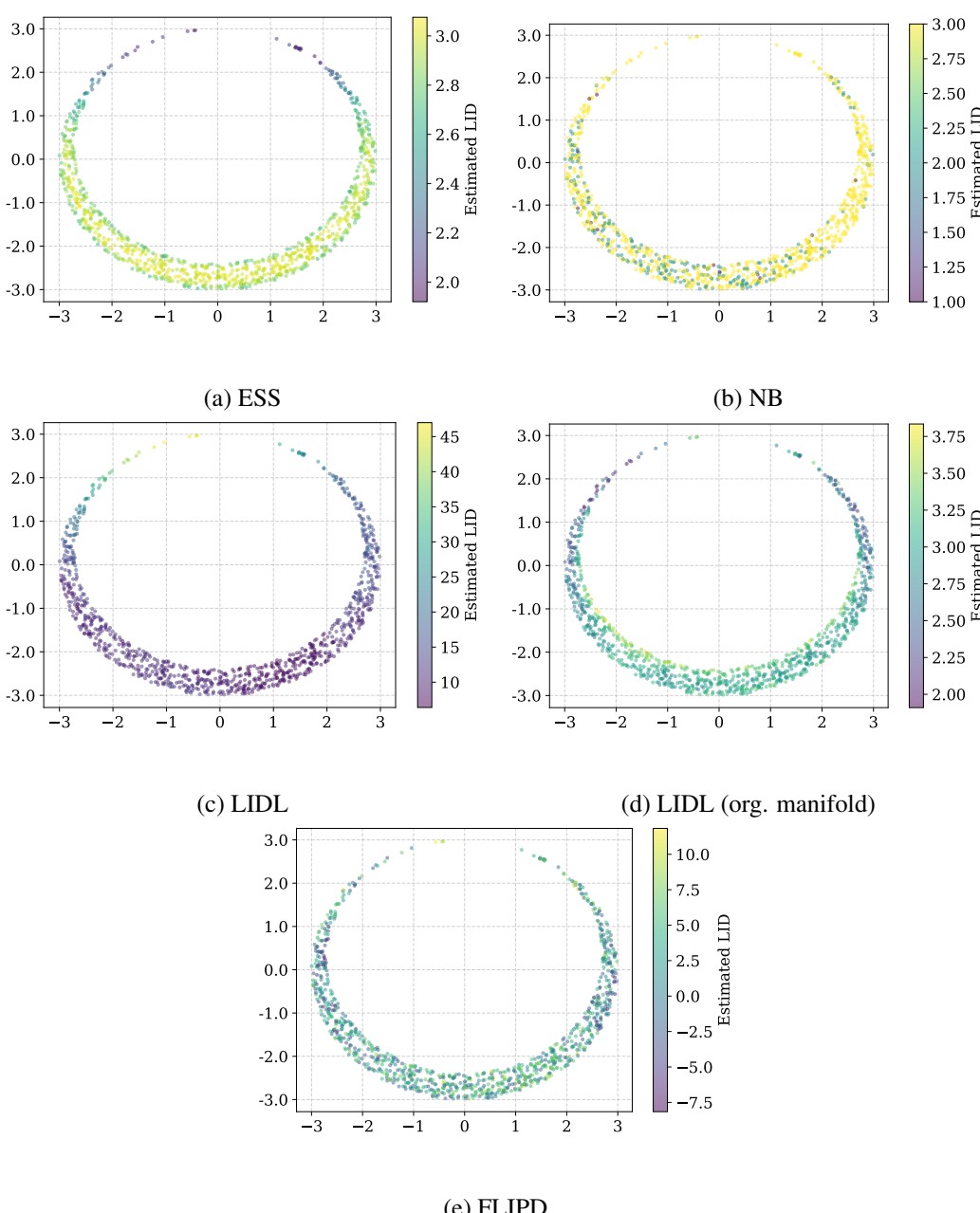

(a) ESS

(b) NB

(c) LIDL

(d) LIDL (org. manifold)

(e) FLIPD

Figure 17: Results for moon dataset for different algorithms. We ranked ESS H due to the accurate estimates for most of the time; We ranked NB and LIDL on original manifold L due to the high variance in the estimate.

## B.5 NEARBY MANIFOLDS

**Funnel**

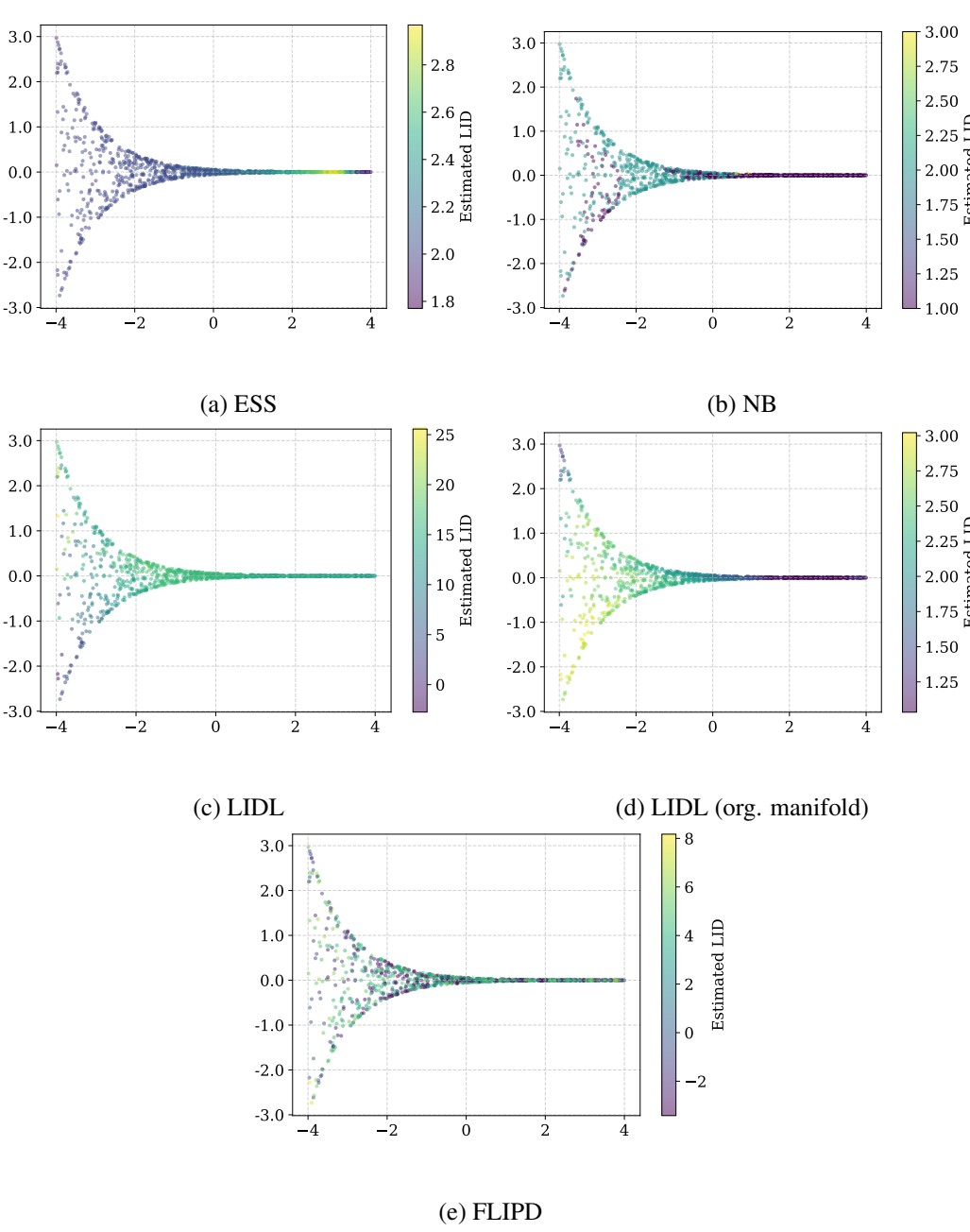

(a) ESS

(b) NB

(c) LIDL

(d) LIDL (org. manifold)

(e) FLIPD

Figure 18: Results for the funnel dataset for different algorithms. We ranked ESS H due to the accurate estimates for most of the time; We ranked NB and LIDL on original manifold L due to the high variance in the estimate.

**Spiral**

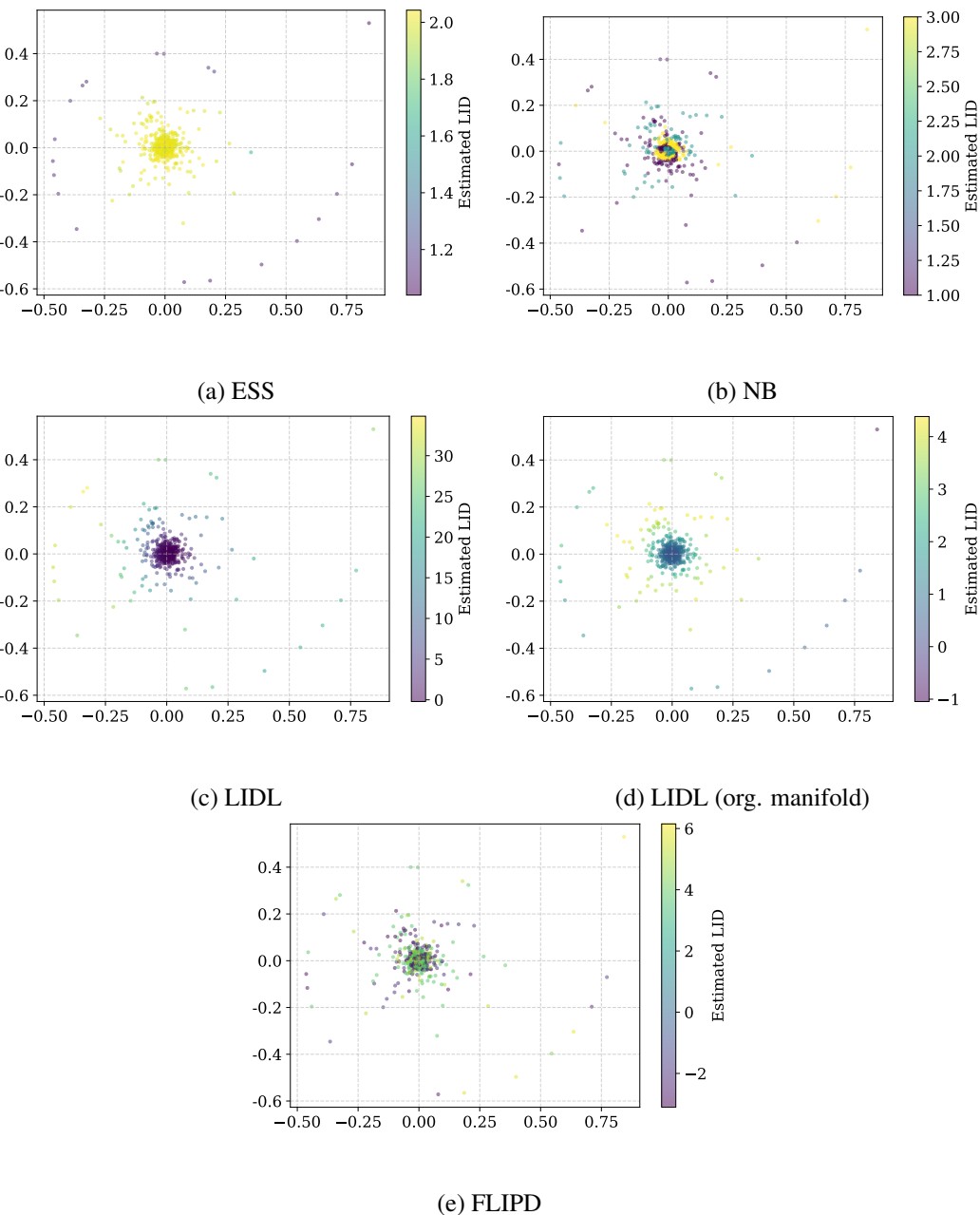

Figure 19: Results for the spiral dataset for different algorithms. We ranked ESS M due to the accurate estimates at the beginning of the spiral; We ranked NB and LIDL on original manifold L due to the high variance in the estimate.

B.6 SAMPLED FMNIST (LID VS SAMPLE SIZE)

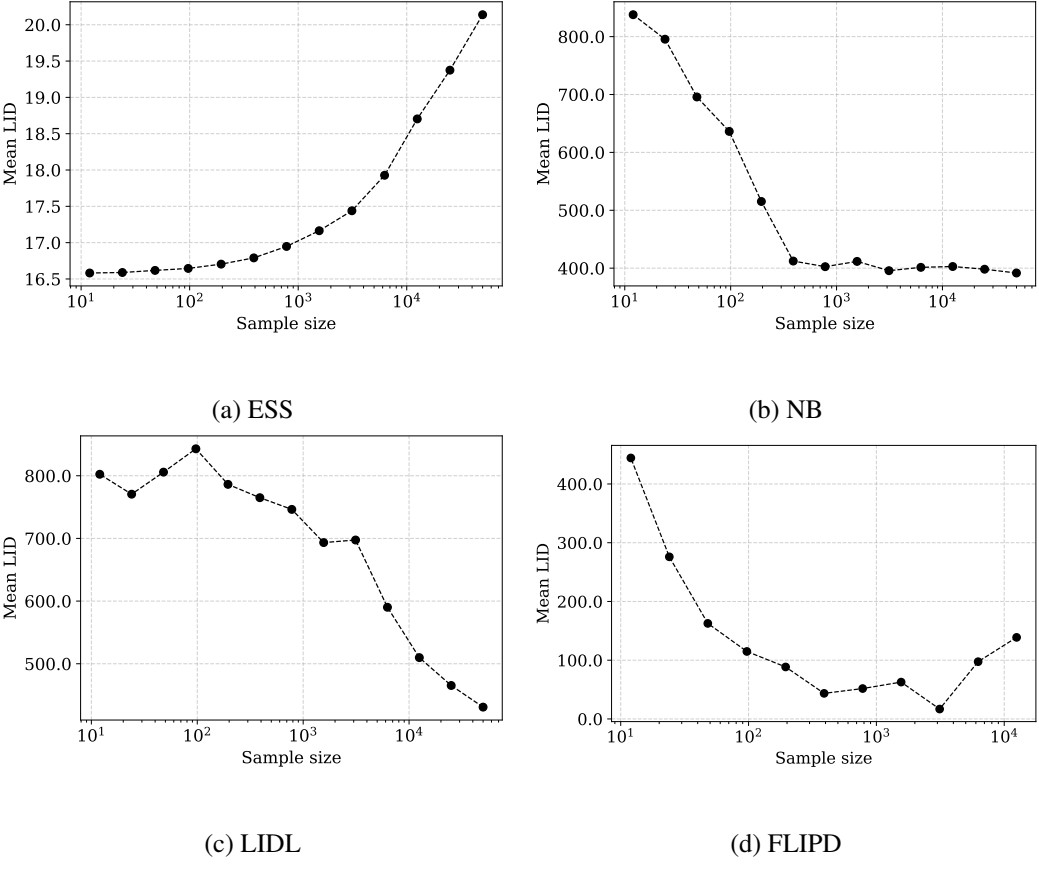

(a) ESS

(b) NB

(c) LIDL

(d) FLIPD

Figure 20: Per-method results for various sample sizes from FMNIST dataset. We ranked ESS L because the estimate dependence on sample size looks worring; We ranked NB H because the estimate stabilizes with sample size; We ranked LIDL L, because the estimate do not stabilize as sample size grows.

## B.7 ESTIMATE INVARIANCE TO DATASET TRANSFORMATION

### B.7.1 ADDING ARTIFICIAL DIMENSIONS

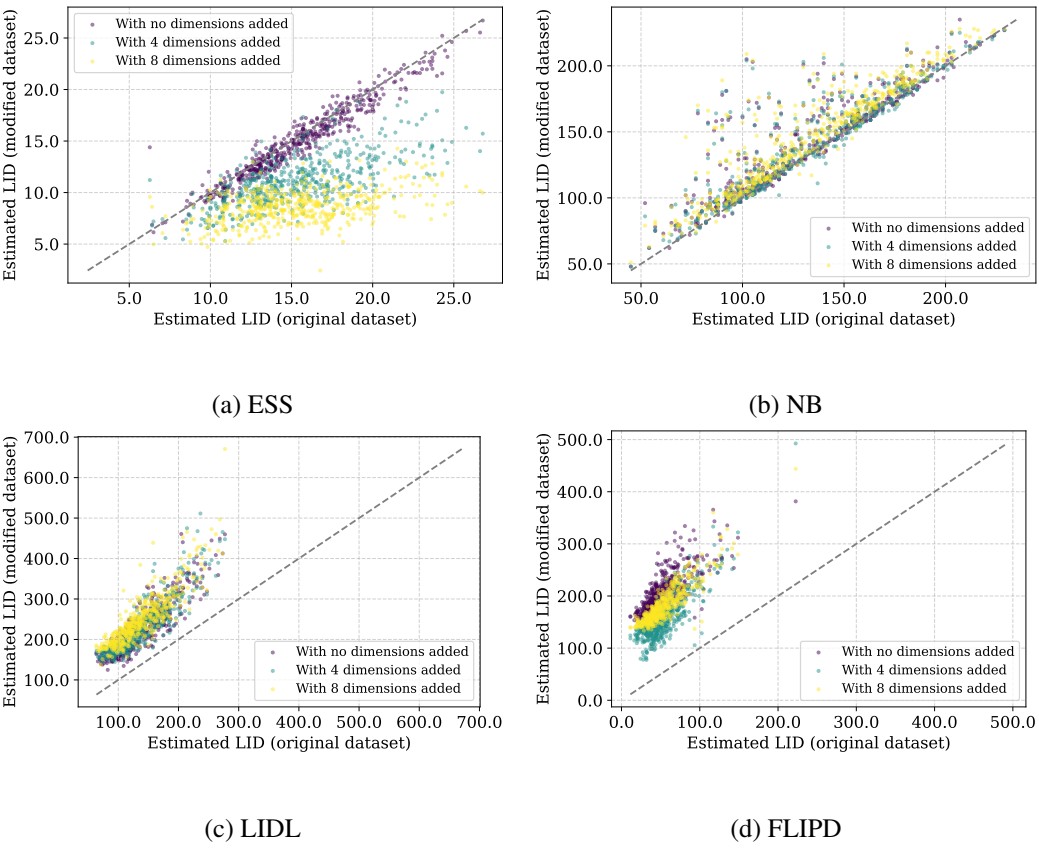

(a) ESS

(b) NB

(c) LIDL

(d) FLIPD

Figure 21: Estimated LID for FMNIST datasets with extra artificial dimensions. Estimates for datasets with added dimensions were calibrated in a way, that number of added dimensions were subtracted from respective estimates, so that on average results should lie on identity line ($y = x$). We ranked NB M because it is the only algorithm that have some points at identity line.

### B.7.2 UPSCALING

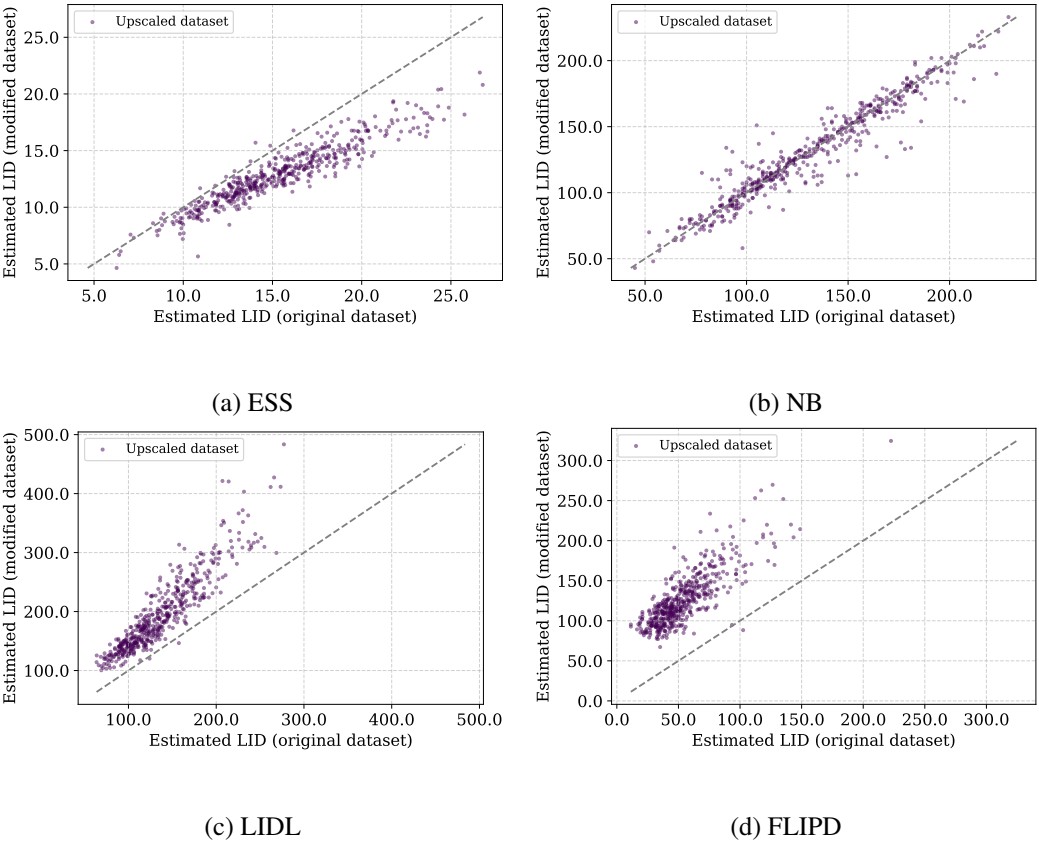

(a) ESS

(b) NB

(c) LIDL

(d) FLIPD

Figure 22: Effect of upscaling on FMNIST data. We ranked NB M because it is the only algorithm that have some points at identity line.

### B.7.3 SPATIAL TRANSFORMATIONS

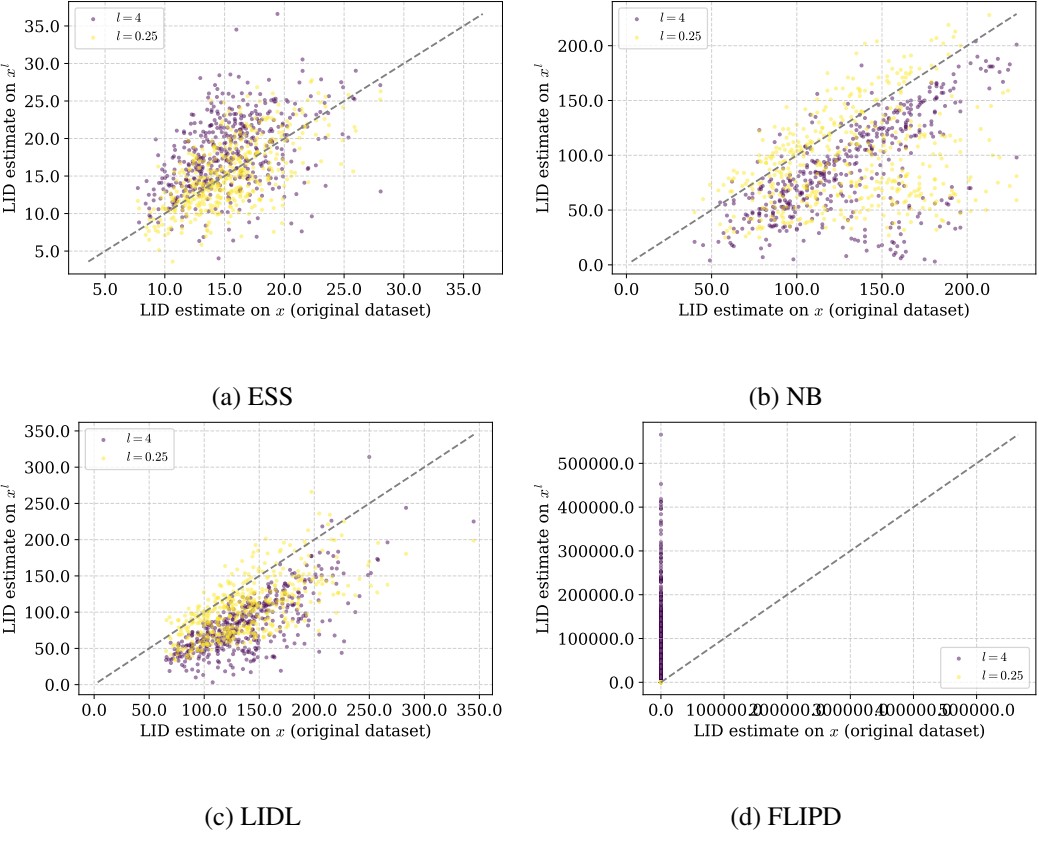

Figure 23: Estimated LID under a FMNIST spatial stretching transformation. Exponent of the transformation is in the legend. We ranked ESS M, because despite the high variance, at least the points were distributed approximately in equal numbers on both sides of the identity line.

## B.8 REAL-LIKE DATASETS WITH KNOWN LID

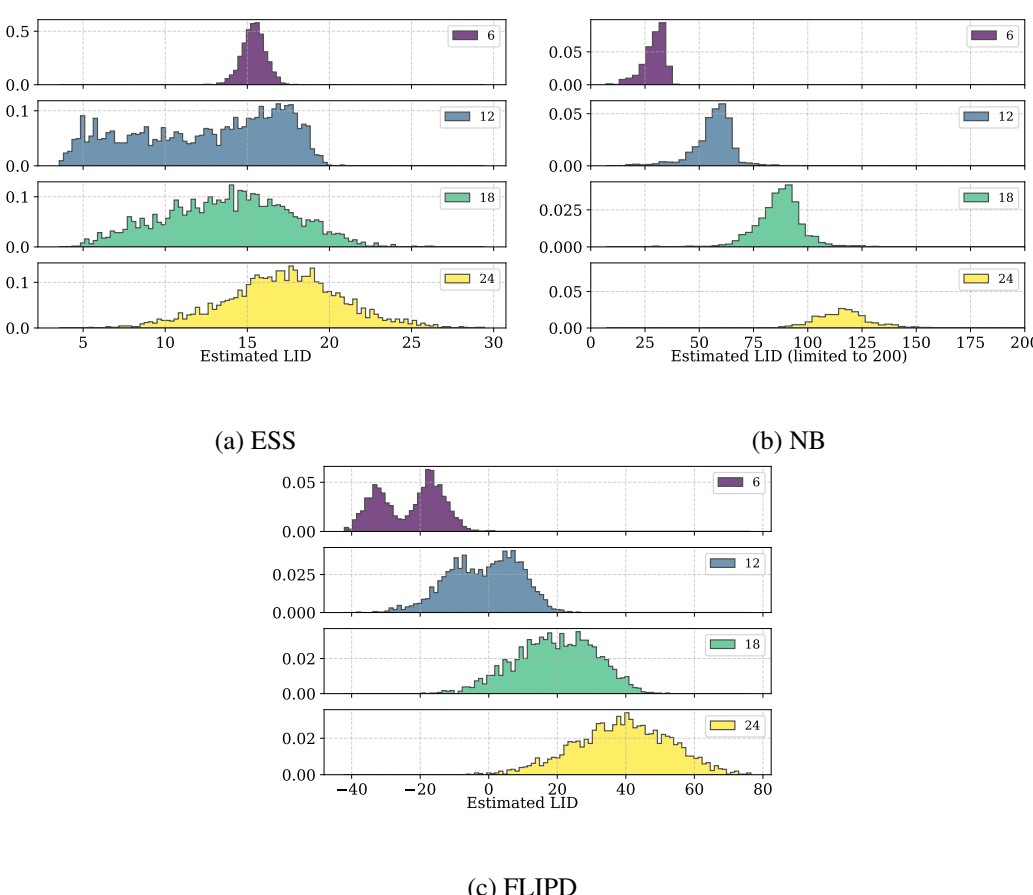

(a) ESS

(b) NB

(c) FLIPD

Figure 24: Estimated LID for arrows dataset. We ranked ESS and FLIPD as L, only because at least some of the estimates are close, but we can see that with high probability it is purely by accident and not because those algorithms are performing well on this task.

## C  RELATED WORK

**LID applications**    There has been a surge of interest in estimating the Local Intrinsic Dimension (LID) of manifolds embedded in higher-dimensional space due to its relevance in various tasks, including representation learning (Ansuini et al., 2019; Li et al., 2018; Rubenstein et al., 2018), dimensionality reduction, and clustering (Vapnik, 2013; Kleindessner and Luxburg, 2015; Camastra and Staiano, 2016; Loaiza-Ganem et al., 2024).  LID estimation is crucial for applications like manifold learning, density estimation (Brehmer and Cranmer, 2020; Caterini et al., 2021; Ross and Cresswell, 2021), and generative auto-encoder models such as VAE (Kingma and Welling, 2014). These methods often rely on a predefined manifold dimension as a hyperparameter, and accurate LID estimation is critical. Rubenstein et al. (2018) demonstrate that errors in estimating the latent space dimensionality can affect the performance of these methods. LID estimation was used as an analytical tool to study the process of training and representation learning in deep neural networks (Li et al., 2018; Ansuini et al., 2019). Results of Pope et al. (2020) reveal that the intrinsic dimension of the dataset impacts the training process of a machine learning model, sample efficiency, and its ability to generalize. Ross et al. (2024) show that LID estimation can be used to investigate memorization in generative models. The earliest methods for estimating manifold dimensionality primarily employed a global approach, as pioneered by Fukunaga and Olsen (1971) and later developed in works such as Minka (2000); Fan et al. (2010).  Recent approaches address the problem locally by analyzing neighborhood geometry in a non-parametric manner Johnsson et al. (2014); Levina and Bickel (2004).

**LID estimation methods**    Over the past decades, numerous methods have been proposed for LID estimation, following various methodologies. Camastra and Staiano (2016) examined the older methods in a comprehensive survey. In our work we test four methods for LID estimation: **ESS** Johnsson et al. (2014), **LIDL**Tempczyk et al. (2022), **NB** Stanczuk et al. (2024), **FLIPD** Kamkari et al. (2024) across the datasets introduced in this paper. We wanted to compare neural methods with older non-parametric one, and ESS was the best performing algorithm among classical methods like MLE (Levina and Bickel, 2004) according to experiments in Tempczyk et al. (2022). We briefly describe tested algorithms in Sec. D.

## D  ALGORITHM DESCRIPTION

**ESS** Johnsson et al. (2014) This method assumes that the data lie in a local neighborhood of a larger set with relatively small curvature and noise. In the ideal case, where there is no noise or curvature, this corresponds to data points being uniformly distributed in a hypersphere. ESS estimates the intrinsic dimension by analyzing the distribution of angles between data vectors. It provides low-bias estimates even when the intrinsic dimension exceeds the number of data points.

**LIDL**Tempczyk et al. (2022) This algorithm uses the rate of change of the probability density under the Wiener process to estimate the local intrinsic dimension (LID). For small diffusion times $t$, the logarithm of the density is a linear function of $\log t$. The slope of this linear relationship is $d - D$, where $d$ is the intrinsic dimension of the manifold and $D$ is the dimension of the ambient space.

**NB** Stanczuk et al. (2024) NB is based on the observation that for small diffusion times, the score function of a diffusion model lies in the normal space of the manifold. Using this fact, authors estimate LID at a given point by applying a singular value decomposition (SVD) to assess the rank of a relevant matrix constructed from the score function.

**FLIPD** Kamkari et al. (2024) FLIPD is a more scalable variant of LIDL, using diffusion models to estimate density.

## E  IDR FORMALIZATION

Take, for example, a dataset $X \subset \mathbb{R}^D$ of face images, and a manifold $M \subseteq \mathbb{R}^d$, already embedded in $\mathbb{R}^d$. If we take $d$ eigenvectors of the covariance matrix of the distribution of face images computed from $X$, their linear combinations will yield a $d$-dimensional space of face-like images in which we can embed a copy of $M$. Sampling from this copy gives us a dataset of face-like images arising from a prescribed manifold.

More formally, suppose we are given a manifold $M$ smoothly embeddable in $\mathbb{R}^d$ through $\phi\colon M \to \mathbb{R}^d$. Given any dataset $X$ embedded in $\mathbb{R}^D$ with $D \geq d$, we may artificially create a dataset diffeomorphic to $M$, interpolating the points of $X$. The procedure starts with computing the mean $\mu_X \in \mathbb{R}^D$ of $X$ and applying PCA to the centered dataset $X - \mu_X$. Then, we take the principal component vectors $u_1, \ldots, u_d \in \mathbb{R}^D$ corresponding to $d$ largest eigenvalues. After that, we embed $M$ in $\mathbb{R}^D$ through $\hat{\phi}(p)\colon M \to \mathbb{R}^D$ given by the formula $\hat{\phi}(p) = \mu_X + \sum_{i=1}^{d} \phi_i(p)u_i$. Finally, we take $\hat{\phi}(M)$ as the new (continuous) dataset, from which we may now sample points.

The embedding $\hat{\phi}$ is just the composition of $\phi$, followed by the embedding of $\mathbb{R}^d$ into $\mathbb{R}^D$, taking the standard basis to the vectors $u_i$. Note that the vectors $u_1, \ldots, u_d$ are unit vectors; therefore, the geometry of the manifold remains unchanged. Nevertheless, the quality of the outcome, measured by visual similarity to the target image domain, is highest when the first $d$ eigenvalues are large, i.e., the original dataset $X$ is not "squeezed" in the corresponding directions.

While the technique is applicable to any target domain, in this paper, we present it for image representation only, as we find it the most insightful and the closest to real datasets on which LID estimation methods have been tested in the past.

In this work we use images from class 7 of FMNIST dataset to fit PCA with $D = 784$. A sample from such dataset is depicted in Figure 1. More discussion on a choice of a dataset to fit PCA can be found in Sec. G.

*Alternative approach to IDR* A possible alternative for the Inverse Domain Representation could utilize the vectors $u_1, \ldots, u_d$ scaled with the corresponding eigenvalues. This approach would ensure that the resulting dataset visually better matches the target image domain; however, it could alter the geometry of the dataset, especially when the first $d$ eigenvalues are significantly different in magnitude. In extreme cases, this could lead to the deformation of the dataset and affect the LID of the transformed data.

For this reason, since preserving the LID of the transformed dataset is crucial for our study, we do not follow this approach. Moreover, the results presented in this paper demonstrate that a transformation that preserves the geometry of the dataset while mapping it to another domain still leads to different LID estimations by neural network-based algorithms, posing a significant challenge for the evaluation of modern methods.

## F    EXPERIMENTAL DETAILS

For all datasets with known dimensionality, we used 100,000 samples for the training set, 1,000 for validation, and 1,000 for test. The only exception was the Arrows (MS) dataset, which uses 100,000/10,000/10,000 split.

**ESS setup**    In our experiments we used ESS implementation from Bac et al. (2021) which can be found under scikit-dimension.readthedocs.io with default hyperparameters if not stated otherwise in the text. Because this implementation cannot calculate LID on unseen data and we can only get predictions for the training set, we have made a decision to jointly train on training and test datasets, but only present the results for the test part of the data.

**NB setup**    For NB experiments, we used the PyTorch implementation along with the environment setup provided by Stanczuk et al. (2024), which is publicly available under github.com/GBATZOLIS/ID-diff. The conducted experiments have been carried out using DDPM Ho et al. (2020) with Adam optimizer ($\alpha = 2\mathrm{e}{-4}$, $\beta = 0.9$, $\epsilon = 1\mathrm{e}{-8}$). The convergence has been assessed using the validation holdout dataset. For a more detailed description of hyperparameters for all conducted experiments (except for Arrows), please refer to MNIST/config.py, which is available in the aforementioned repository. The only hyperparameter adjustments we made were connected to the shape of the input data. For the Arrows experiment, we used based our config on the celebA/ddpm.py file.

**LIDL setup**    To perform experiments with LIDL, we utilised the official PyTorch implementation (github.com/opium-sh/lidl) released by Tempczyk et al. (2022). All the experiments (except the ones on the original manifold and padding) has been performed on MAE Papamakarios et al. (2017)

with 5 layers and 5 hidden units. For the version with the original manifold and padding, we had to went down with the number of layers to 4 due to unfavourable scalability and performance constraints. Each experiment has been perofrmed with the diffent set of $\delta$ triplets, such that $\delta \in \left\{ \left( 2^{-(n-1)}, 2^{-n}, 2^{-(n+1)} \right) | n \in \{1, \ldots, 7\} \right\}$. The rest of the hyperparameters remained in line with the version presented by Tempczyk et al. (2022).

**FLIPD setup** To obtain the results presented in this article we utilized a fork from August 12, 2024, of the repository listed under github.com/layer6ai-labs/flipd by the authors of FLIPD Kamkari et al. (2024). As for now authors recommend utilizing github.com/layer6ai-labs/dgm_geometry for LID estimation experiments. For all datasets, the architecture of the diffusion model was the same up to input layer dimensionality. We used the same MLP architecture for diffusion models as authors which is described in Section C appendix of FLIPD article Kamkari et al. (2024). For all datasets we ran 1000 epochs of training for each network, choosing the best model measured by its' validation loss. For the datasets with known LID values we chose $t$ which yielded lowest MAE error given all samples from test set. For datasets with unknown LID values we used heuristic presented with original article with the caveat that we searched for the *knee* for values of $t$ larger than the maximum average estimated LID. We did it as for many of the datasets for certain t values instabilities occurred which resulted in extreme LID estimates. Those values prevented the kneed package from performing correctly as it assumes monotonicity of the function.

Table 5: Approximate duration for model training on RTX 2080 Ti GPUS.

| Dataset | NB | FLIPD | LIDL |
|---|---|---|---|
| FMNIST (upscaled) | 1d @ 2GPU | 5.5h @ 2 GPU | 9h @ 1 GPU |
| FMNIST (downscaled) | 0.5d @ 2GPU | 4h @ 2 GPU | 6h @ 1 GPU |
| FMNIST (stretched $x^4$) | 1.5d @ 2GPU | 4.5h @ 2 GPU | 7.5h @ 1 GPU |
| FMNIST (stretched $x^{0.25}$) | 2d @ 2GPU | 4.5h @ 2 GPU | 7.5h @ 1 GPU |
| FMNIST (add dim, +4d) | 2d @ 2GPU | 5h @ 2 GPU | 8h @ 1 GPU |
| FMNIST (add dim, +8d) | 4d @ 2GPU | 6h @ 2 GPU | 10h @ 1 GPU |
| Spiral | 5.5d @ 2GPU | 10.5h @ 2 GPU | 22h @ 1 GPU |
| Uniform | 3.5d @ 2GPU | 6h @ 2 GPU | 9.5h @ 1 GPU |
| Funnel | 4.5d @ 2GPU | 6h @ 2 GPU | 9.5h @ 1 GPU |
| Moon | 5d @ 2GPU | 6.5h @ 2 GPU | 10h @ 1 GPU |
| Gaussians | 1d @ 2GPU | 7.5h @ 2 GPU | 11h @ 1 GPU |
| Spheres | 1d @ 2GPU | 7h @ 2 GPU | 10h @ 1 GPU |
| Spaghetti | 1.5d @ 2GPU | 6h @ 2 GPU | 9h @ 1 GPU |
| Arrows | 10d @ 8GPU | 10h @ 2 GPU | – |

## G   DISCUSSION & IMPORTANT REMARKS

During the development of the article, we received numerous helpful questions and suggestions from the scientific community, for which we are grateful. As many of them led to an interesting discussion and remarks, we publish an excerpt of them in this section.

**Validity of the IDR transformation.** Our claims regarding the properties of IDR are supported by the theory behind principal component analysis. Since PCA is an affine transformation that only rotates the manifold and translates it within the ambient space, it does not alter the key geometric properties we consider. For example, this transformation preserves distances between points and does not affect curvature – both of which are crucial factors in many of our experiments.

**Why did we choose the 7th class from FMNIST for IDR?** When we apply PCA to most of the image datasets, the resulting representations no longer resemble meaningful images. They often appear as chaotic blobs of light and dark, lacking coherent structure. This occurs because applying PCA to a complex manifold does not guarantee that the resulting distribution for first $n$ vectors will be Gaussian. While PCA centers and rotates the data, the resulting distribution often contains holes. If PCA is used in reverse as a generative model – i.e., by sampling from a standard normal distribution in the reduced space and applying the inverse PCA transformation-samples drawn from these holes can lead to unnatural and noisy outputs. This is precisely why we chose a specific shoe class from FMNIST: it was one of the few datasets where the data was sufficiently dense in the first 20 PCA components to yield visually coherent and plausible images. In contrast to other classes, this particular FMNIST category produced sharper edges and more meaningful shapes, making it suitable for our experiments in the image domain. Another reason we did not test other classes was due to computational budget constraints. Training on datasets with higher dimensionality, especially RGB images instead of grayscale, would be several times more expensive – an overhead we could not accommodate within our current resources.

**Why do we assume that LID is a homogenous integer?** In this paper, we adopt a definition of LID that aligns with those used in recent neural-based LID estimation algorithms we analysed. While the assumption of integer-valued LID is a meaningful limitation that deserves future attention, we observe that current algorithms still struggle even with relatively simple manifolds under this assumption. Addressing these remains our current focus. That said, our IDR dataset generation approach is capable of embedding any manifold into the data space while preserving its structure, making it suitable for generating data with non-integer dimensionalities manifolds with singularities when we generate one as an input.

Our work focuses specifically on the LID estimation problem. This perspective may not capture all the complexities relevant to downstream applications in representation learning, some of which may benefit more from global dimensionality measures. However, there is clear and ongoing interest in local LID estimation. Our goal is to support this active line of research by offering targeted benchmarks that reveal key weaknesses in existing estimators, many of which, as our results show, struggle on specific, controlled instances. While the broader issues of non-integer and heterogeneous LID values are important, addressing them comprehensively lies beyond the scope of this work. Nevertheless, we believe our contributions serve as a useful and timely step toward improving future methods. Finally, we note that although many current estimators assume local homogeneity of density, our aim is not to critique accepted algorithms but to provide tools that help the community develop more robust approaches. It is also worth mentioning that some methods, such as LIDL, are already capable of handling non-uniform densities to some extent (Tempczyk et al., 2025).

**Fourier-based alternative to PCA.** An anonymous reviewer suggested using Fourier or DCT bases, which is indeed very interesting. They suggested to consider a 2D discrete Fourier expansion of an $N \times N$ image in a form of

$$X(i,j) = \sum_{u,v=0}^{N-1} \hat{X}(u,v) e^{2\pi i (u_i + v_j)/N}, (i,j) \in \{0, ..., N-1\}^2$$

and retain bases with a coefficients having a high $\hat{X}(u,v)$, which can proxy the top PCA directions for a smoother, frequency-oriented chart.

We agree such approach seem to be suited to the specific case of image datasets. However, the framework suggested by the reviewer is also more complex. What we need is a single $d$-dimensional

orthonormal system for the whole dataset, which is the case, e.g., in the PCA approach. In the proposed approach, we need to choose a subset of a larger orthonormal basis $\mathcal{B}$, given something we may roughly describe as "compatibility score" for each sample from the dataset and each basis vector (the coefficient in the expansion of the sample in basis $\mathcal{B}$). Then there is an additional step of aggregating this information to obtain a single orthonormal system. Given the expansions $x = \sum_{b \in \mathcal{B}} \langle x, b \rangle b$ of every in our dataset, and a fixed $b \in \mathcal{B}$ we could look at the functions $x \mapsto \langle x, b \rangle$ and try to use their norms, e.g. $\ell^1$ or $\ell^2$ norms. We could also use these coefficients to perform a voting on basis vectors. Nevertheless, despite discussed shortcomings, IDR was enough to show underperformance of LID algorithms, which was one of our goals.

**Why the Arrows dataset (MS) is the most challenging one?** This dataset is more challenging for existing algorithms because the manifold has some nasty properties that others presented don't. E.g. our theoretical considerations show that the manifold has many V-shape corners as an artifact of translating and rotating the arrow on the image. This may pose a challenge for the existing algorithms designed with the simpler manifold shapes in mind.

**What is the underlying manifold in Arrows dataset, what the parameterization looks like, and how the transformation preserves the intended ID?** In the Arrows dataset, when two arrows overlap, their color vectors are added without clipping, leading to unbounded RGB values, but preventing information loss. An image of a single arrow can be parametrized by the manifold $M = [0,1]^5 \times S^1$, where the first five dimensions correspond to (normalized) 3 components of RGB color and 2 coordinates of the center of the arrow. The $S^1$ factor corresponds to the rotation of the arrow. For $k$ arrows, the image is parametrized by $M^k$, and so there is a mapping $f \colon M^k \mapsto A$, where $A$ is the space of all $k$-arrow images. This mapping is not a bijection, but is close enough to a bijection for LID purposes. That is, outside of a union of lower-dimensional submanifolds of $M^k$, all the coordinates are distinct, i.e. every arrow has a different color, position, and angle. What we claim is that the restriction of $f$ to this set is a smooth bijection with its image.

**How does local density in the funnel manifold affect LID estimation?** We acknowledge that variations in local density can indeed affect the final LID estimate, as demonstrated in our experiments on sample size (Figure 9). However, our observation is that higher local density generally reduces the likelihood of the algorithm misclassifying a lower-dimensional manifold as a higher-dimensional one. When points from different manifolds are located close together, the algorithm may incorrectly treat them as part of the same manifold, leading to an overestimation of intrinsic dimensionality. In this experiment, our goal was to show that different algorithms vary in the distance at which this misclassification occurs – the smaller this distance, the better the algorithm.

**Practical guidance for LID estimation** In practice, for low-dimensional manifolds we recommend using ESS, as it is fast and stable, though it becomes biased in higher dimensions. If the datasets are higher-dimensional and one has more computational resources, we recommend using NB. It showed the best stability and performance and is not biased in higher dimensions. Although the training time is not much longer than for other neural algorithms, the prediction time is an order of magnitude higher than, e.g., LIDL, but the performance is much better than any other algorithm we tested.

# H  SOME OTHER RESULTS FOR LIDL AND FLIPD FROM THE COMPARISON

Mean LID estimated by LIDL for diffetent values of δ

| benchmark | [0.0039 0.0078 0.0156] | [0.0078 0.0156 0.0312] | [0.0156 0.0312 0.0625] | [0.0312 0.0625 0.125 ] | [0.0625 0.125 0.25 ] | [0.125 0.25 0.5 ] | [0.25 0.5 1. ] |
|---|---|---|---|---|---|---|---|
| Spiral (IDR) | 195.2 | 38.1 | 26.677 | 1.9763 | 5.7654 | 2.1493 | -14.271 |
| Uniform (IDR) | 89.149 | 42.094 | 33.343 | 34.703 | 29.495 | 28.05 | 23.904 |
| Funnel (IDR) | 27.524 | 27.831 | 19.581 | 14.176 | 18.466 | 12.414 | -2.7025 |
| Moon (IDR) | 26.38 | 20.263 | 22.344 | 12.314 | 23.193 | 14.268 | -2.9294 |
| Gaussians (IDR) | 86.434 | 22.523 | 15.501 | 17.243 | 25.48 | 14.003 | -0.043722 |
| Spaghetti (IDR) | 72.248 | 31.989 | 31.24 | 10.254 | 6.9001 | 1.3064 | -3.972 |
| Spheres (IDR) | 40.56 | 19.827 | 20.471 | 17.899 | 12.726 | 18.274 | 11.025 |

deltas

Figure 25: On this plot we present how values of average LID estimate changes for different values of δ parameter and different IDR datasets for LIDL algorithm.

# I  USE OF LARGE LANGUAGE MODELS (LLMS)

LLMs were used only for light copy-editing (e.g., grammar and phrasing) after the technical content was written by the authors. All content was verified by the authors.

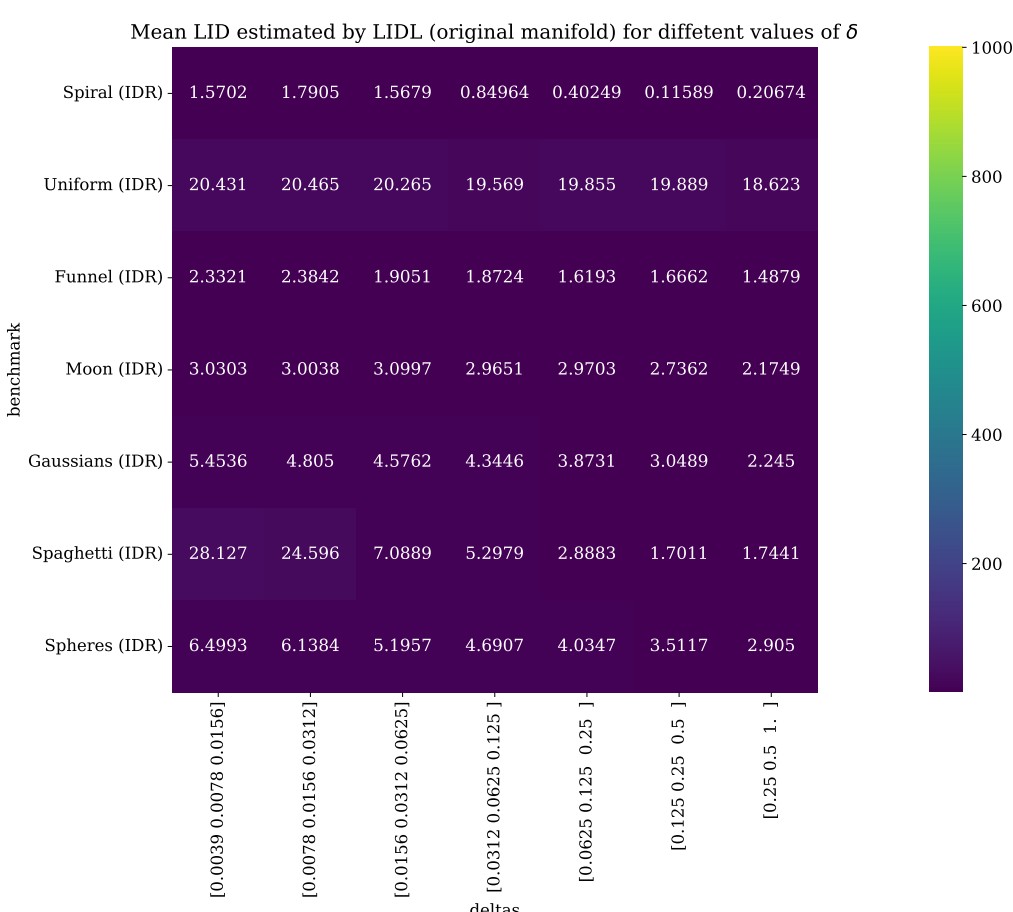

Figure 26: On this plot we present how values of average LID estimate changes for different values of $\delta$ parameter and different datasets for LIDL algorithm. Those datasets are made of original manifold coordinates before IDR transformation. Ambient space of those datasets is 30.

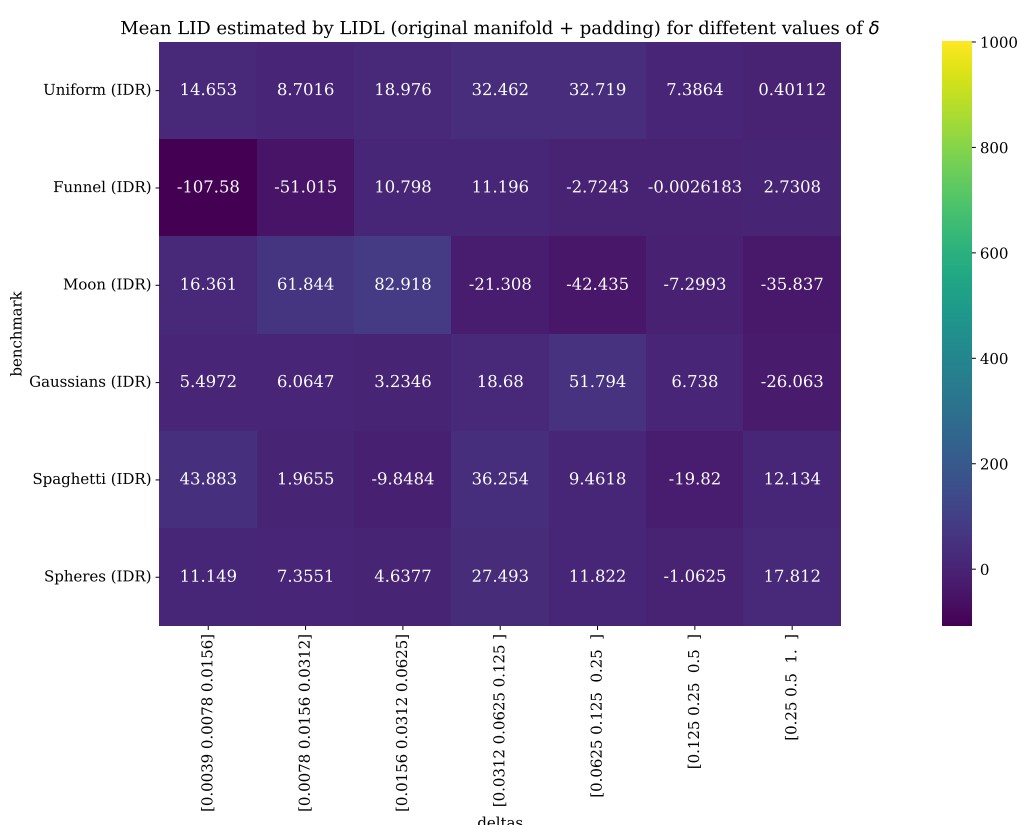

Figure 27: On this plot we present how values of average LID estimate changes for different values of $\delta$ parameter and different datasets for LIDL algorithm. Those datasets are made of original manifold coordinates before IDR transformation padded with 0 to be of higher dimension. Ambient space of those datasets is 784.

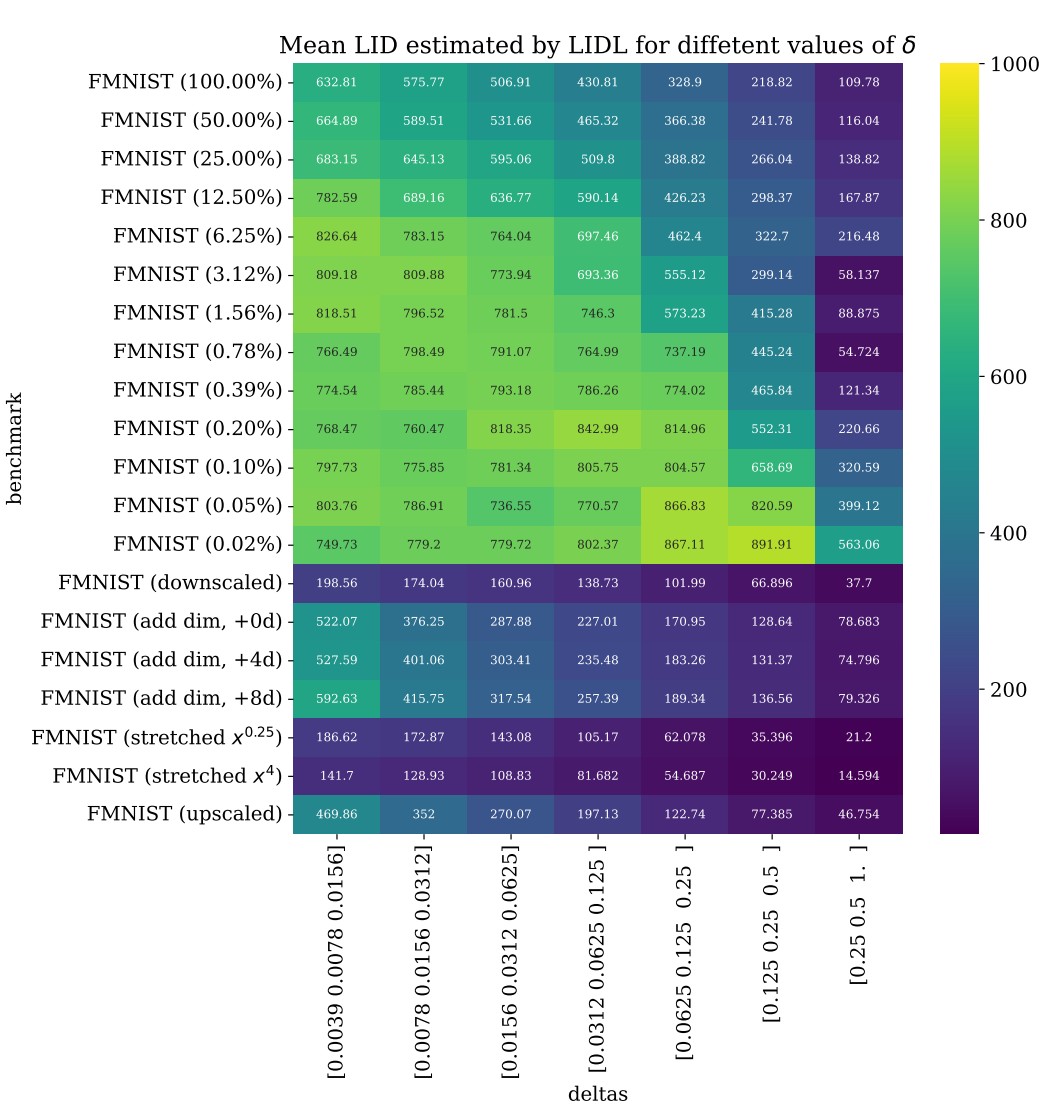

Figure 28: On this plot we present how values of average LID estimate changes for different values of $\delta$ parameter and different modified FMNIST datasets for LIDL algorithm.

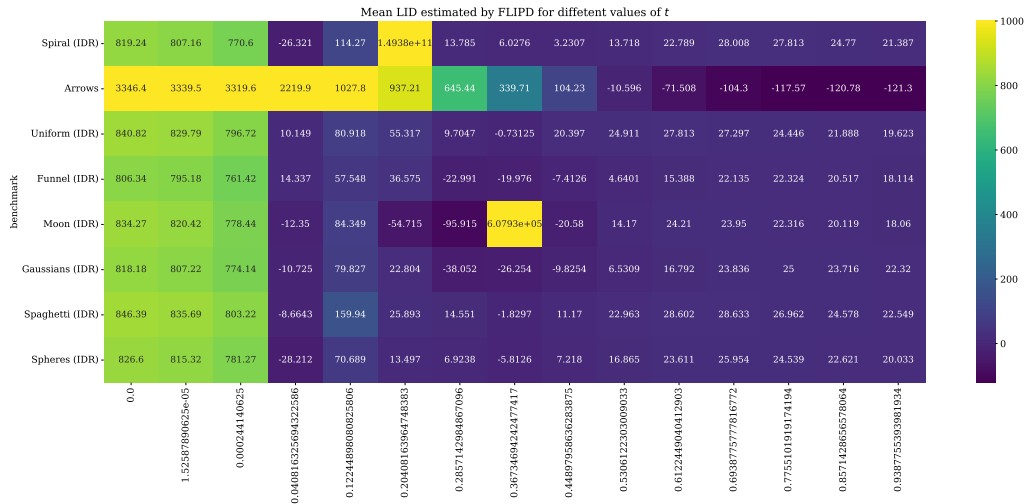

Figure 29: On this plot we present how values of average LID estimate changes for different values of $t$ parameter and different IDR datasets for FLIPD algorithm. To improve clarity, we present results for every 4th $t$.

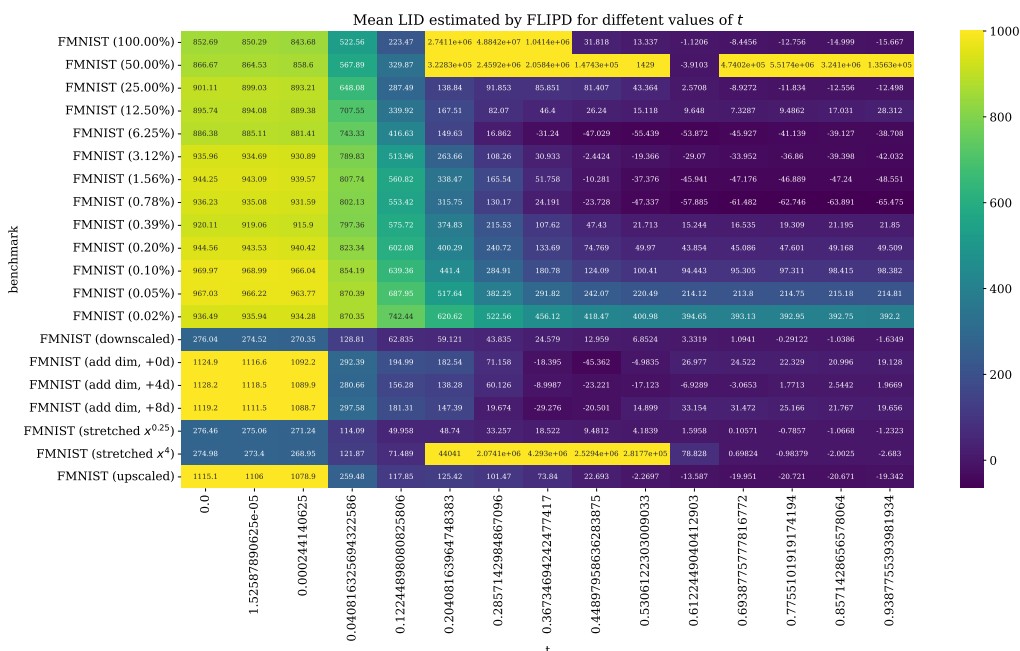

Figure 30: On this plot we present how values of average LID estimate changes for different values of $t$ parameter and different modified FMNIST datasets for FLIPD algorithm. To improve clarity, we present results for every 4th $t$.

