# OpenReview forum: "Why We Need New Benchmarks for Local Intrinsic Dimension Estimation"
_ICLR.cc/2026/Conference — ICLR 2026 Poster_

### Official Review · Reviewer_QG1L · 2025-10-23

**Soundness:** 2
**Presentation:** 2
**Contribution:** 3
**Rating:** 2
**Confidence:** 4

**Summary:**

This paper proposes several methods for generating high-dimensional datasets with known intrinsic dimensions from various perspectives, for the purpose of evaluating local intrinsic dimension (LID) estimation methods, and presents examples of how these datasets can be used to assess the performance of different LID estimation algorithms.

**Strengths:**

The paper proposes several methods for generating datasets that are well-suited or particularly challenging for intrinsic dimension estimation from various perspectives, and it conducts comprehensive evaluation experiments on representative (neural) LID estimation methods.

The diverse data manifolds presented in Subsections 3.1 through 3.9 each represent scenarios where successful intrinsic dimension estimation is desirable, and as such, the proposed benchmarks provide a valuable contribution in their current form.

**Weaknesses:**

The paper emphasizes the importance of benchmarking LID estimation and proposes various methods for constructing datasets; however, it lacks a discussion that would convince the broad ICLR audience why (local) intrinsic dimension estimation is important and to what extent it is a significant problem in the first place.

While the procedures for creating benchmark datasets are described in sufficient detail, the discussion of LID estimation methods themselves is insufficient. The appendix provides only brief, one-paragraph summaries of individual methods, but it does not offer insights into which methods succeed or fail on each benchmark dataset, and why, which would be necessary to derive meaningful understanding from the experiments.

**Questions:**

As pointed out in the “Weaknesses” section, the paper should clearly articulate why (local) intrinsic dimension estimation is important in the first place.

In addition, to convey the significance of the proposed benchmark datasets, it is necessary to explain why neural network–based methods are important, and simultaneously, to discuss the limitations of traditional fractal-dimension approaches such as box-counting dimension.

Specifically, please describe what the limitations of non–neural-based methods are, and clarify why the proposed benchmark datasets are designed primarily for evaluating neural network–based methods.

---

> ### Author Response · Authors · 2025-11-19
>
> First of all, we would like to thank the reviewer for the time and effort invested, as well as for the constructive and valuable feedback. We are pleased to address all the concerns raised.
>
> Below, we provide some examples of LID applications:
>
> Representation learning and training dynamics in NN: Layerwise LID traces how internal features contract or reorganize during training; such dimensional collapse signals abstraction and helps diagnose under/overparameterization [1–3].
>
> Dimensionality reduction: LID offers a data-driven criterion for choosing projection targets: low-LID regions can be safely compressed, while heterogeneity across classes argues against a single global dimensionality [4–7].
>
> Manifold and density estimation: In flow, and score-based models, LID validates the assumed intrinsic dimension and reveals when learned densities deviate from the data manifold. Variations in local density or score rank across noise scales expose curvature or thickness effects [8–10].
>
> Latent size in autoencoders: For VAEs and related models, aligning latent dimensionality with measured LID improves reconstruction and stability, while mismatch may degrade sample quality [11–13].
>
> Intrinsic dimension correlates with learning efficiency: low-LID regions yield smoother optimization and better generalization, whereas high-LID zones mark geometrically hard regions needing more capacity or stronger inductive bias [3].
>
> Memorization in generative models: High-LID regions often coincide with memorization, offering a geometric view of overfitting and privacy risk; LID-aware regularization or data curation can mitigate these effects [14].
>
> Union-of-manifolds hypothesis: Real data are better described as unions of manifolds of differing dimensions. LID reveals dimension shifts between components (e.g., categories, textures) and helps test this assumption in image domains [15].
>
> Out-of-distribution detection: OOD samples usually lie in regions of expanded local dimension. Incorporating LID improves separability beyond likelihood scoring and enhances robustness under covariate shift [16].
>
> Reconstruction and uncertainty diagnostics: Reconstruction error in VAE correlates with local LID, high-LID samples are harder to reconstruct, making it a model-agnostic indicator of difficulty. Similarly LID can approximate predictive uncertainty of classifiers by flagging input regions with high LID [17].
>
> Why neural network–based methods are important & what are the limitations of non–neural-based methods?
>
> The main problem with non-neural methods are that they all fail to deliver accurate and unbiased estimates in higher-dimensional setting as shown in Table 1 & 2 and Figure 7 in [17]. Authors show that best of those methods (ESS) can have relative error ~30% and all other methods (like Correlation Dimension or MLE) have errors between 70-90%. The neural methods are currently the only methods that can achieve much higher accuracy on these datasets (at the level of few % as shown in [17]).
>
> Finally, we would like to point out that many of the cited results, including the NN-based LID estimation methods, appeared in top-tier venues (ICML, ICLR, NeurIPS, AAAI), confirming the community's interest in the area.
>
> [1] Alessio Ansuini et al. Intrinsic dimension of data representations in deep neural networks. NeurIPS 2019
>
> [2] Chunyuan Li et al. Measuring the intrinsic dimension of objective landscapes. ICLR 2018
>
> [3] Phil Pope et al. The intrinsic dimension of images and its impact on learning. ICLR 2020
>
> [4] Vladimir Vapnik. The Nature of Statistical Learning Theory. Springer Science & Business Media, 2013
>
> [5] Matthäus Kleindessner et al. Dimensionality estimation without distances. In Artificial Intelligence and Statistics, 2015
>
> [6] Francesco Camastra et al. Intrinsic dimension estimation: Advances and open problems. Information Sciences 2016
>
> [7] Gabriel Loaiza-Ganem et al. Deep generative models through the lens of the manifold hypothesis: A survey and new connections. arXiv
>
> [8] Johann Brehmer et al. Flows for simultaneous manifold learning and density estimation. 2020
>
> [9] Anthony L. Caterini et al. Rectangular Flows for Manifold Learning. NeurIPS 2021
>
> [10] Brendan Leigh Ross et al. Tractable Density Estimation on Learned Manifolds with Conformal Embedding Flows. NeurIPS, 2021
>
> [11] Paul K. Rubenstein et al. On the latent space of Wasserstein auto-encoders. arXiv
>
> [12] D. P. Kingma et al. Auto-encoding variational Bayes.
>
> [13] Ilya Tolstikhin et al. Wasserstein auto-encoders. ICLR 2018
>
> [14] Brendan Leigh Ross et al. A geometric framework for understanding memorization in generative models, 2025
>
> [15] Bradley C. A. Brown et al. Verifying the union of manifolds hypothesis for image data. ICLR 2022
>
> [16] Hamidreza Kamkari et al. A geometric view of data complexity: Efficient local intrinsic dimension estimation with diffusion models. ICML 2024
>
> [17] Tempczyk et al. LIDL:Local Intrinsic dimension using approximate likelihood, ICML 2022

---

### Official Review · Reviewer_iGTt · 2025-10-28

**Soundness:** 3
**Presentation:** 3
**Contribution:** 3
**Rating:** 6
**Confidence:** 3

**Summary:**

The paper argues that current evaluations of local intrinsic dimension (LID) estimators are either too simple (synthetics with known LID) or unverifiable (real data with unknown LID). It proposes a benchmarking framework that: (i) maps the same manifold across domains (e.g., to images/audio) to test architecture/inductive-bias invariance; (ii) hardens existing datasets along specific geometric/measure aspects (non-uniform density, curvature, boundaries, nearby/thin manifolds); and (iii) introduces controlled real-data transformations (monotonic warps, ambient-dimension extension, auxiliary dimension injection) that induce known LID shifts or preserve LID, enabling relative ground-truth checks on real data. Across a suite of synthetic, “real-like,” and transformed FMNIST datasets, the authors show that several recent neural LID estimators (LIDL, FLIPD, NB) exhibit systematic failures under these stressors, whereas a classic method (ESS) is often more stable on low-dimensional cases; none is universally robust.

**Strengths:**

Originality.

Introduces a domain-mapping toolbox (IDR) to test the same manifold across architectures/domains—surfacing inductive-bias sensitivity that simple synthetics miss.

Designs targeted stressors (non-uniform densities, curvature, edges, nearby/thin manifolds) and controlled real-data transforms (ME/ASE/ADI) to create falsifiable checks (LID preserved or shifted by a known amount).

Quality.

Broad, well-motivated experimental suite: Gaussians/Spheres/Uniform/Spaghetti, Moon/Funnel/Spiral (nearby/thin), Arrows (real-like with known LID), plus FMNIST transforms; summarizes per-aspect outcomes in tables/figures.

Identifies concrete failure modes (e.g., NB overestimation on Gaussians/Spheres; LIDL/FLIPD instability; sample-size bias).

Clarity.

Clear “what/why” in Introduction and contributions; formalization of IDR (PCA-based embedding that preserves manifold geometry for the tests) and per-transform rationale are well explained; figures are readable and tied to claims.

Significance.

Benchmarks uncover architecture/domain sensitivity and lack of invariance that prior evaluations hide; they provide a useful bar for future LID methods and highlight where current neural estimators fail (e.g., Arrows, boundaries, size-dependence).

**Weaknesses:**

1. Despite the stated “any continuous domain” framing, experiments are concentrated in the image domain (FMNIST, Arrows) and four LID estimators. This limits generality of the conclusions about cross-domain invariance. Add at least one non-image domain (audio/EEG) to exercise ME/ASE/ADI beyond images.

2. For ME/ASE/ADI, the paper argues LID preservation/shift qualitatively; however, formal guarantees (and error bounds under finite samples) are not given. Tighten the theory or at least provide simulation checks where the induced ΔLID is analytically known before porting to FMNIST.

3. IDR relies on PCA directions from a single FMNIST class chosen for visual coherence; the Discussion acknowledges this choice and compute constraints. This may bias domain mappings and downstream conclusions about architecture invariance. Consider validating IDR with alternative bases (e.g., DCT) or multiple classes, and quantify any distortions with pairwise-distance/curvature diagnostics.

4. The authors note that more principled metrics per aspect are “future work.” Even simple aggregated scores (MAE vs. known LID; calibration of ΔLID; invariance indices) with confidence intervals across seeds would make comparisons more decisive.

5. Several methods are sensitive (e.g., ESS neighbors; LIDL δ; FLIPD t). The paper shows this qualitatively, but a uniform protocol (grid, selection rule, robustness bands) would avoid accusations of cherry-picking and clarify practical deployability.

**Questions:**

1. For ME/ASE/ADI, can you provide proof sketches or lemmas that (a) ME preserves LID; (b) ASE preserves LID; (c) ADI increases LID by the number of injected independent parameters plus synthetic verifications before FMNIST?

2. Beyond the PCA argument, can you report quantitative diagnostics (isometry error, curvature change) showing IDR preserves manifold geometry for your choices; and try one alternate basis (e.g., DCT) to check the invariance of conclusions?

3. Could you include one audio benchmark (e.g., monotonic resampling—ASE; dynamic range warp—ME) to substantiate the “any continuous domain” claim?

4. Will you add per-aspect scalar metrics with 95% CIs (e.g., MAE/ΔLID error, invariance score) to complement plots, and report seeds and tuning protocols for each method?

5. Arrows seems especially challenging; can you ablate which property (corners, occlusion, color composition) breaks each method, and whether smoothing/anti-aliasing mitigates it?

---

> ### Author Response · Authors · 2025-11-19
>
> First of all, we would like to thank the reviewer for the time and effort invested, as well as for the constructive and valuable feedback. We are pleased to address all the concerns raised:
>
> Q1: For ME/ASE/ADI…
>
> These results follow from general theory asserting invariance of dimension under well-behaved mappings. For instance, in ME, we use monotonic continuous maps, and the whole mapping we create is a homeomorphic embedding. Preservation of LID then follows from invariance of topological dimension under homeomorphisms. If we restrict ourselves to smooth maps with non-zero derivatives, we can use invariance of differential dimension under smooth embeddings. In the other methods we can similarly assert that what we actually do is applying a sufficiently regular map to our manifold. Assuming smoothness (we required only continuity in the paper, but in fact used smooth functions, and this simplifies the arguments), in ASE we embed re-embed $M$ using a map of the form $\mathbb{R}^d \ni x -> (x, F(x))$ which has a smooth inverse - the projection onto the first $d$ dimensions, and thus preserves the dimension. In ADI, which we didn’t formalize, we apply maps that have non-degenerate Jacobians, so they are immersions and preserve the local dimensions of their domains. The increase of dimension is determined by the dimension of the domain of the new embedding we define.
>
>
> Q2: Beyond the PCA argument…
>
> To speak about isometry error and curvature changes, M needs to have more structure e.g. a Riemannian metric. While in general we do not require such structure, we often work with Riemannian manifolds. Notice that the transformation we apply to M can be expressed as a composition $\tau \circ \psi \circ \phi$, where:
> $\phi\colon M \to \mathbb{R}^d$ is the embedding of $M$ into euclidean space that we assume is given to us and
> $\psi\colon \mathbb{R}^d \to \mathbb{R}^D$ is a linear map taking the standard basis of $\mathbb{R}^d$ to principal component vectors $u_i$
> $\tau(x) = x + \mu_{X}$ is a translation of $\mathbb{R}^D$
> Since the principal component vectors are orthogonal and have unit length, the mapping $\psi$ is an isometric embedding. The translation $\tau$ is an isometry, and so the composition $\tau \circ \psi$ preserves distances and curvature exactly. Any distortions come from $\phi$, which is arbitrary. On the other hand, if we assume that the structure of $M$ is induced from the embedding (thus making $\phi$ an isometry by assumption), then the whole composition is an isometry, preserving the metric and curvature exactly. These conclusions hold for any orthonormal basis.
>
> Q3: Could you include…
>
> Because of the page limit restrictions we wanted to focus on one domain, and we chose images because it is the only domain used by the LID estimation community so far. This makes it much easier to compare our results with other papers. If we would include one audio benchmark we would have to drop one image benchmarks, which would make our paper less focused and incomplete in this domain, so while it is a great idea to test it on audio, we argue that it deserves separate paper judging on the size of the experiments section in our paper.
>
>
> Q4: Will you add…
>
> We would be more than happy to calculate CI, this would require us doing at least 4 times more computation than we used so far. We cannot afford that at this point with our academic resources. We didn't fix any seeds at any point in our training, but we will be happy to add more information you requested to the appendix section with experimental details.
>
> Q5: Arrows seems…
>
> Could you please explain what you mean by ablating corners in the Arrows dataset? Occlusion occurs in a few % of datapoints and it does not occur in the single-arrow images. If this would be a problem this should not affect the overall LID estimate for most of the dataset, and we see that there are no points with correct estimates for any of the data points for any algorithm.
>
>
> “IDR relies on PCA….”
> We thank the Reviewer for an interesting future direction of work. It definitely can guide follow up research when more sophisticated methods, than the existing ones, are challenged. In this paper we managed to pinpoint major issues of the methods with a simple transformation.
>
> “The authors note that…”
>
> We believe that some of those or similar metrics can be found in Tables 2,3,4 in Appendix A. As stated in our reply to Q4, calculating CI was outside of our academic computational budget.
>
>
> “Several methods are sensitive…”
>
> We agree with that. Unfortunately you could not do grid over the parameters with a dataset with unknown ground truth about LID. The authors of the method should give a method to calibrate algorithms for new datasets. And because none of the tested algorithms do this in the paper, we asked  the authors of those methods to provide us with a set of universal parameters or took the parameters used on the image datasets from the paper itself.

---

### Official Review · Reviewer_Sef9 · 2025-10-31

**Soundness:** 4
**Presentation:** 3
**Contribution:** 3
**Rating:** 8
**Confidence:** 4

**Summary:**

The paper analyzes algorithms for estimating local intrinsic dimension, focusing on ESS, NB, LIDL, and FLIPD. The experiments cover simple synthetic datasets with known dimension and realistic datasets with complex structure and unknown dimension. For the realistic case, the paper applies transformations that preserve manifold dimension or change it in a known way and then measures changes in reported dimension to evaluate robustness. The study investigates controlled factors that affect LID estimates, including manifold curvature, thickness, sample size, non uniform density, and proximity to boundaries.

**Strengths:**

- The experimental design explicitly probes factors known to affect LID estimates, including curvature, thickness, sample size, non uniform density, and proximity to boundaries.

- The experimental section is thorough.

- The empirical design spans simple known dimension synthetic data and complex unknown dimension real data, which provides a broad stress test for the methods.

- The paper systematically highlights shortcomings of existing LID methods and existing benchmarks through controlled experiments.

**Weaknesses:**

- The Gaussian (IDR) experiment does not report the number of samples used, and tail regions such as standard deviation beyond 2 standard deviations require high sample sizes to get enough samples in the region to have stable estimates, especially for model trained methods like FLIPD and LIDL. The number of training and test samples should be reported across other experiments as well.

- The Funnel (IDR) experiment is placed under nearby manifolds, but as described it is a single component and seems more relevant to thin manifolds.

- The Spiral (IDR) experiment under nearby manifolds appears to have extremely low point density on the outer spiral, which can cause any algorithm to fail.

- Section 3.5 nearby manifolds uses terms like distance to the neighbor on the same manifold versus distance to the nearest neighbor in the ambient space without a precise definition, and the Funnel and Spiral experiments do not make these distances clear.

- Figure 9 uses a log x axis and allocates a large fraction of the range to fewer than 100 samples, which is hard to interpret for FMNIST where the ambient dimension is 28*28.

- Figure 23 (d) would be clearer with the x axis limited to a lower range because the current figure is unreadable.

- The paper mentions using PyTorch interpolation for resizing in Upscaled ASE and possibly other experiments, and known aliasing artifacts in this method might affect LID [1].

- Section 2 mentions audio but there are no audio experiments, and removing the audio discussion would improve focus.

- Section 3.7 cites work showing ESS is invariant to sample size for artificial datasets, but comparable results for the other algorithms on synthetic data where ground truth is known are missing.

[1] Parmar, Gaurav, Richard Zhang, and Jun-Yan Zhu. "On aliased resizing and surprising subtleties in gan evaluation." Proceedings of the IEEE/CVF Conference on Computer Vision and Pattern Recognition. 2022.

**Questions:**

- What is the number of samples used in the Gaussian IDR experiment, and is it sufficient for reliable estimation in the tails? What are the number of samples used for the other experiments?

- Can the authors define precisely distance to the neighbor on the same manifold versus distance to the nearest neighbor in the ambient space, and explain how these distances are instantiated in the Funnel and Spiral experiments?

- In line 1471, it is claimed that "Arrows (MS) has many V-shaped corners as artifact of translating and rotating". Can this be further explained?

- When two arrows overlap it is claimed that RGB values are added (Line 1471), but this effect is not visible in Figure 2. Can the authors clarify this?

- Minor typos:

  - Line 141: "Sec. 1" should be "Sec. 2."

  - Line 168: It is unclear whether the manifold is S^4 or embedded in a 6 dimensional space. Please clarify the intended statement.

  - Line 242: The set description contains typos. Please correct the notation.

  - Line 312: "algorithm error" should be "algorithm's error."

  - Line 317: "Figure 8" refers to another paper but links to the Figure 8 in the current paper.

  - Line 412: "too big our computational" should be "too big for our computational."

  - Line 445: "worse to" should be "worse than."

  - Line 455: The word "while" should be removed.

  - Figure references are inconsistent, alternating between "Fig." and "Figure" throughout the paper.

---

> ### Author Response · Authors · 2025-11-19
>
> First of all, we would like to thank the reviewer for the time and effort invested, as well as for the constructive and valuable feedback. Thank you for spotting typos, we will fix all of them in the revised manuscript. We are pleased to address all the other concerns raised:
>
> Q1: What is the number of…
>
> Thanks. Indeed, the manuscript is not detailing the sample size. We will post and clarify these numbers in the revised version. For all datasets with known dimensionality, we used 100k samples for training, 1k for validation, 1k for test. The exception was the arrows dataset: 100k/10k/10k. In each of the gaussians there are ~150 points that are further away than 2.5 sigma from the mean of the distribution.
>
> Q2: Can the authors define…
>
> By distance on the manifold we mean the geodesic distance: the length of the shortest path that stays on the manifold. The ambient distance is the straight-line (Euclidean) distance. Imagine a U-shaped wire (manifold): put A and B at the two tips and C at the bottom of U. On-manifold, A’s nearest neighbor is C (shortest path along the wire); in ambient space, A’s nearest neighbor is B (the tips are close in a straight line).
> Q3: In line 1471…
> Thank you for pointing this out. We intended to claim that the dataset is sampled from a manifold with corners, which is maybe not such a common terminology in ML. While manifold looks locally like Euclidean space, and manifold with boundary looks like a half-space (boundary points of the manifold have neighborhoods modelled on the boundary plane of the half-space), manifold with corners is modelled on a $d$-dimensional cube, admitting corners of different dimensions. Since the coordinates and RGB values are bounded, the parameter space is a product of a multidimensional cube and circles (corresponding to rotation angles) - parameters in lower-dimensional faces of this cube yield corner points of the manifold.
>
> Q4: When two arrows overlap…
>
> Thank you for pointing this out. This is the problem with Figure 2 itself. The Figure comes from the earlier experimental setup where the overlapping arrows were overwriting the values for non-black pixels. This was changed in later versions of the experiments. In the revised version, we will adjust the figure accordingly.
>
> “Section 2 mentions audio… “
>
> The reviewer makes a valid argument. Our goal was to emphasize that the proposed dataset-generation techniques could be extended to the audio domain, although doing so would exceed the capacity of this manuscript. For the sake of space limitations, we decided to focus on the canonical image domain in this work. That said, for the camera-ready version we will revise the phrasing regarding the audio dataset to improve clarity and focus, without diminishing the message about the generality of the proposed techniques.
>
> “Section 3.7 cites…”
>
> We observed that, when comparing the results of our experiments on FMNIST with the sample-size experiments on synthetic datasets reported in the original LIDL paper, FMNIST is more challenging for LIDL in terms of sample size. Since this paper focuses on performance on real-world datasets, which we found to be more challenging at the same time, we therefore concentrate our sample-size analysis on real-world data only.
>
> “The Funnel (IDR)...”
>
> This is a single manifold but folded in a way that opposite parts of the manifold are getting closer to each other as we move along the first dimension , so the points from other parts of the manifold are affecting the LID estimate.
>
> “The Spiral (IDR)...”
>
> What you see on the Figure is the test set, which is 100 times less dense than the training set.
>
>
>
> “Figure 9 uses …”
>
> There is a very important distinction to be made here: the ambient space can be high-dimensional, and this is not problematic as long as its dimensionality is lower than the sample size. We do not know the exact dimensionality of different parts of the FMNIST data manifold, and we observed that, even for very small sample sizes, there is a very consistent trend across points in all the curves, so we decided to show this plot (as an interesting result) for the whole range of sample sizes.
>
>
> “Figure 23 (d) would be clearer…”
>
> Our point is that FLIPD is far from the identity line. If we cap the x-axis at the maximum observed x, the identity line would lie close to the x-axis, and the plot could appear (misleadingly) to follow the identity line at first glance.
>
> “The paper mentions…”
>
> We thank the Reviewer for pointing this out. We do use bilinear interpolation in several experiments. We ran sanity checks on the transformed data (e.g., verifying Gaussian and Uniform after IDR) and computed PCA to see whether LID could be inferred; it worked as expected. These artifacts therefore did not materially affect the manifold’s topological structure to any observable extent.

---

> > ### Comment · Reviewer_Sef9 · 2025-11-26
> >
> > I thank the authors for their detailed and thoughtful responses.
> >
> > - In my initial review I forgot to mention that having a slight discussion regarding the runtime of various algorithms seems relevant (as pointed out by reviewer 5JNw).
> > - For the final point about clean resizing. I am not sure about the exact sanity checks performed but theoretically, this function (increasing the size of the image) is an immersion so the resulting manifold will have the same dimension. However, the aliasing artifacts might cause some of the methods to fail (which itself is of interest). Having a simple experiment with a clean resize can behave as ablating those effects.
> >
> > I recommended acceptance in my initial review, and I maintain this recommendation.

---

> > > ### Author Response · Authors · 2025-12-04
> > >
> > > As the discussion period comes to an end, we would like to thank the Reviewer for their work and the valuable discussion.

---

### Official Review · Reviewer_5JNw · 2025-11-01

**Soundness:** 4
**Presentation:** 3
**Contribution:** 3
**Rating:** 8
**Confidence:** 3

**Summary:**

This work is seeks to bridge a gap in common benchmarking approaches for LID estimators: the literature typically focuses on either (1) simple manifolds of known dimension or (2) realistic manifolds of unknown dimension. To address this gap, a toolkit is proposed for designing new LID benchmarks whose (1) geometric properties are understood but which (2) resemble realistic data. Foremost among these tools is inverse domain representation (IDR), wherein a simple, well-understood manifold $M$ is embedded isometrically into the space of a real-world dataset $X$ using the principal components of $X$. In the paper, this amounts to creating images that *look* like FMNIST but have the geometric properties of spheres, curves, solid hypercubes, etc. Other tools are also introduced, e.g. warping the manifold or ways of adding ambient dimensions.

The work constructs a sequence of datasets by applying these tools to simple known manifolds with geometric properties that are pathological (at least from the perspective of common LID estimators). Four estimators are compared on these datasets - expected simplex skewness (ESS), normal bundle (NB), LIDL, and FLIPD. These estimators, and especially the latter 3 neural net-based ones, underperform compared to results on simpler datasets in their respective papers, establishing this set of benchmarks as a potential new frontier for LID estimation.

**Strengths:**

- In my opinion, this work does what it sets out to do. It goes a decent way towards bridging the aforementioned gap between simple LID benchmarks and real-world datasets.
- It is also clearly written.
- Its strength is in the way it systematizes methods for creating benchmarks and applies them to create new ones. Its "toolbox" is clever and original. If it gains traction, I could see it forming a new de facto leaderboard for LID estimators.
- Its main weakness is when it starts to work with individual estimators - see weaknesses for thorough feedback on this.

Nevertheless, as I said, this work achieves what it sets out to do, and I recommend it for acceptance.

**Weaknesses:**

Much of this work ends up being a direct comparison of 4 popular LID estimators. I think the work's main weak spots are in the specifics of this comparison:
- The work only summarizes at a high-level the results of its benchmarks, and provides almost no analysis of *why* specific LID estimators do well or poorly on the constructed benchmarks. The community is left by themselves to figure out what exactly is going on with the individual estimators. I believe this work would be more impactful if it were to propose fewer datasets but provide more explanation of the failure modes of specific estimators on each.
- Pursuant to the previous point, this work suggests no improvements nor provides clear next steps for improving the state of the art. Having experimented with these methods on a vast suite of LID estimation problems, the authors should be in a good position to recommend solutions to the highlighted failure modes.
- In my opinion, there is some nuance missing in the way the work compares estimators:
	- Efficiency goes ignored in this analysis. My understanding is that, if we were to choose the fastest possible usable hyperparameters for each method and amortize training time, FLIPD > LIDL >> NB >> ESS in terms of runtime cost. This is approximately the opposite order to the ranking in this manuscript.
	- The discussion work mostly treats these estimators as drop-in replacements for each other when in fact they each have advantages beyond error and efficiency. In particular, NB/LIDL/FLIPD are useful (1) online, for new points and (2) when the model's own LID is of interest. Some more specific advantages:
		- NB supplies normal vectors of the manifold, which have utility beyond a simple LID number.
		- LIDL is the only of these that works for normalizing flows and density estimators in general.
		- The FLIPD estimate of a point is differentiable and can be estimated in a single forward pass of its model.
		If perfect LIDs were my only consideration, ESS would always be my default choice as long as it is tractable (but it is often not).

**Questions:**

- See weaknesses 1 and 2. Do you have any insights on why the individual estimators perform as they do in specific scenarios and how they can be improved?
- Can you expand on these phrases and perhaps provide a specific citation if appropriate? I had understood density constancy as a requirement more for classical LID estimators than newer neural ones like NB, LIDL, FLIPD.
	- "in LIDL this bias is a function of the laplacian of the density"
	- "existing algorithms make specific assumptions during derivation -- [...] local density constancy"

---

> ### Author Response · Authors · 2025-11-19
>
> First of all, we would like to thank the reviewer for the time and effort invested, as well as for the constructive and valuable feedback. We are pleased to address all the concerns raised:
>
> Q1: See weaknesses 1 and 2…
>
> The reviewer raised a very valid argument, namely that further analysis of why specific LID estimators behave the way they do is a very interesting research direction and would make an additional contribution to the paper. The main goal of the paper was to present an extensive set of benchmarks that pinpoint various problematic cases for modern LID estimation methods. Focusing on the completeness of this list of problematic instances showcases fundamental shortcomings of the methods and highlights the need for deeper analysis. While we fully agree that such analysis is necessary, the role of this paper is to convince the community of this fact. Further analysis explaining the behavior of the methods on these benchmarks is the subject of our current work and, due to its depth, will require a separate manuscript. Nevertheless, we are happy to provide reviewer with our preliminary intuitions.
>
> In the case of LIDL, the results from [1] suggest that the mathematical part of the algorithm is solid and the main source of error is the imperfect density estimate. For instance, the results for delta going to 0, should provably give correct LID, however it does not approach the true LID, but goes towards the dimensionality of the ambient space. Similar case for the FLIPD, where additionally the algorithm lacks a very precise algorithm to read the density from the plots with “knee”. The NB algorithm is the hardest one to analyze because there is no deeper mathematical analysis of its behavior as for LIDL and FLIPD in [1], so it's hard to tell which part of the algorithm may be responsible for its unsatisfactory behavior. Our angle of attack for this problem first focuses on doing mathematical analysis of the algorithm in a similar way to LIDL analysis in [1].
>
>
> Q2: Can you expand…
>
>  "in LIDL this bias is a function of the laplacian...":
>
> This is shown in [1] in Eq. (11).
>
> "existing algorithms make specific assumptions ...":
>
> Thanks for pointing this out. The phrasing is unfortunate because it suggests the assumption was used in the proofs; however, for example, in the proof of the LIDL algorithm there is a limiting argument that takes the delta parameter (t in [1]) to 0. In practice, we cannot make delta/t (or the diffusion step in NB) arbitrarily close to 0, and in that regime we typically assume that, locally (at the scale of delta/t or the diffusion step), the manifold and the density are approximately flat, so that the algorithm remains valid even when the scale parameter cannot be taken infinitesimally small. In this case we have to assume that locally (in terms of delta or diffusion step) the manifold and density is approximately flat to be able to assume that the algorithm still works when we cannot go with our scale parameter infinitesimally close to 0. As an example the only case that works for all delta/t in LIDL is uniform density on the euclidean space as shown in [1] Eq (13). We will modify this sentence to make that clear in the camera-ready version of the paper.
>
> “nuance missing in the way the work compares estimators”:
>
> As we were primarily concerned with the estimated LID, we didn’t include complexity analysis. However, we agree that this information could be important for numerous reasons, including practical ones. In general, the runtime cost order is in line with Reviewer’s observations.
> ESS, while being model- and parameter-free, suffers from being neighbour-based, which effectively makes it the slowest inference-time algorithm in practical, high dimensional settings.
> For remaining ones, we need to factor in the training cost. LIDL with normalizing flows requires training multiple models, whereas both FLIPD and NB require a single model.
> Most importantly, the inference time for FLIPD is fastest - citing its authors,“FLIPD provides a massive speedup over the NB estimator -- $\Theta(kF)$ vs.\ $\Theta(KF + D^{2}K)$ -- especially in high dimensions where $K > D \gg k$”.
>
> “drop-in replacements for each other” :
>
> That is a very interesting perspective on LID estimation methods. While they provide varying quality in estimating LID, they also offer additional information that goes beyond a single intrinsic dimension value. Importantly, we note that all the considered methods were originally designed with the goal of effectively estimating LID, and this original purpose is what we benchmark in this paper. We believe that the additional characteristics of specific algorithms and the extra information about tested manifolds they output can be very useful for understanding the behavior of the methods and may lead to potential improvements on the benchmark instances provided in this paper.
>
> [1] Tempczyk et al. Wiener process perspective on LID estimation methods, AAAI 2025

---

### Author Response · Authors · 2025-12-04

As the discussion period comes to an end, we would like to thank all the Reviewers for their work and the valuable discussion.

---

### Meta-Review · Area_Chair_yAZJ · 2026-01-05

**Summary:**

Reviewers mainly pointed out:
- Lack of explanation of why specific LID estimators succeed or fail or clear next steps (e.g. 5JNw).
-  A range of clarifications on the theoretical guarantees or justifications of the proposed transformations (e.g. iGTt)
- Technical details in the experiments and results (e.g. Sef9 - sample size and density, iGTt - IDR bias and hyperparameter selection)

**Reviewer Concerns:**

The authors provided adequate clarifications and rebuttals for the technical questions.

This leaves the lack of a "why" or next steps, which the authors essentially argue is out of scope or out of reach.  While reasonable, this limits the value of the paper and results for a broad audience.

**Reviewer Scores:**

Reviews were already favorable, starting with 8/8/6/2.  Following the discussion, I would expect a final score slightly above 6.

The authors are encouraged to make the story more accessible to a broader audience, highlighting immediate implications for practical considerations of LID estimation and its impact on specific applications.  As it stands, it seems this discussion would be of interest to a narrow segment of the audience who are actively working on LID estimation -- much less for other researchers or practitioners.

For example, it would be great to say: based on our analysis, the common practice of using method-X for application-Y likely results in lower/higher LID estimates leading to performance issues A/B/C.  Perhaps there can be a recommendation to instead use method-Y, or follow a slightly more involved process to decide between the estimates from method-X and method-Y.  Better if the authors can include such a demonstration -- and how the insights gained by this work help ameliorate those issues.

---

### Decision · Program_Chairs · 2026-01-26

Accept (Poster)